



# Linked thick to thin – skinned inversion in the central Kirthar Fold Belt of Pakistan

Ralph Hinsch[1], Chloé Asmar[1], Muhammad Nasim[2], Muhammad Asif Abbas[2], Shaista Sultan[2]

[1]OMV Upstream, Exploration, Vienna, 1020, Austria

[2]OMV (Pakistan) Exploration GmbH (subsidiary of United Energy Group Limited), Islamabad, Pakistan

**Abstract**

The Kirthar Fold Belt is part of the lateral collision zone in Pakistan linking the Makran accretionary wedge with the Himalaya orogeny. The region is deforming very obliquely, nearly parallel to the regional S-N plate motion vector, indicating strong strain partitioning. In the central Kirthar Fold Belt, folds trend roughly N-S and their structural control is poorly understood.

In this study, we use newly acquired 2D seismic data with pre-stack depth migration, published focal mechanisms, surface and subsurface geological data as well as structural modelling with restoration and balancing to constrain the structural architecture and kinematics of the Kirthar Fold Belt.

The central Kirthar Fold Belt is controlled by Pliocene to recent inversion of Mesozoic rift related normal faults. Focal mechanisms indicate dip-slip faulting on roughly N-S trending faults with angles in the order of 45°, which are too steep for

newly initiated thrust faults. The hinterland of the study area is primarily dominated by strike slip faulting. The inverting faults do not break straight through the thick sedimentary column of the post-rift and flexural foreland; rather the inversion movements link with a series of detachment horizons in the sedimentary cover, progressively imbricating the former footwall of the normal fault. Due to the presence of a thick incompetent upper unit (Eocene Ghazij shales) these imbricates develop as passive roof duplexes. Finally, the youngest footwall shortcut links with a major detachment and the deformation propagates

to the deformation front, forming a large fault-propagation fold. Shortening within the studied sections is calculated to be on the order of 20%.

The central Kirthar fold belt is a genuine example of hybrid thick- and thin-skinned system in which the paleogeography controls the deformation. The locations and sizes of the former rift faults controls the location and orientation of the major folds. The complex tectonostratigraphy (rift, post rift, flexural foreland) alone with the strong E-W gradients defines the

mechanical stratigraphy, which in turn controls the complex thin-skinned deformation.

## 1. Introduction

The external regions of fold-thrust belts are typically interpreted using templates from classical thin-skinned thrust related deformation. However, more and more thrust belts are interpreted as showing a strong influence of linked basement involved deformation (Lacombe and Bellahsen, 2016 and references therein). The inversion of inherited rift faults is one possibility of





thick-skinned contribution in a thrust belt. Furthermore, the direct linkage of inverting basement faults to thin-skinned thrusts in the external parts of orogens (hybrid thick- and thin-skinned system) have received more attention recently (e.g. Giambiagi et al., 2008; Fuentes et al., 2016; Mahoney et al., 2017).

As an outcome of structural investigations for hydrocarbon exploration we are able to report about a well constrained example

of linked thin-skinned to thick-skinned deformation at the deformation front of the central Kirthar Fold Belt in Pakistan. This example illustrates the kinematical linkage between inverting deep reaching faults and the associated thin-skinned deformation, and also shows how this hybrid system is strongly controlled by its paleostructural (extension/rift) and paleogeographic (mechanical stratigraphy) inheritance.

The Kirthar fold belt belongs to the western fold belts in Pakistan which is a zone of strike-slip faults and fold belts along the

western lateral boundary of the Indian plate linking the Himalaya orogen with the Makran accretionary wedge (Lawrence et al., 1981, Bannert et al., 1992, Fig. 1). The northern Kirthar Fold Belt as well as the Sulaiman Ranges to the North of it, had been traditionally interpreted with classical fold-thrust belt geometries with an implied shortening magnitude of 30-40% (Banks and Warburton, 1986; Humayon et al., 1991; Jadoon et al., 1992, 1993, see location examples in Fig. 1). In contrast, the southern Kirthar Fold Belt had been field investigated (Smewing et al., 2002a) and modeled (Fowler et al., 2004) with the

conclusion that the deformation in the belt is dominated by inversion tectonics with an estimated shortening of approximately 17%.

The reported contrasting styles of deformation in the Kirthar fold belt would imply significant and potentially implausible along-strike variation in the shortening magnitudes. In order to understand how deformation is accommodated along this lateral plate boundary, adequate estimations on shortening are essential. Furthermore, constraining the deformation style is

fundamentally important for the exploration of resources, as seismic interpreters usually use template structural models to interpret in areas of poor seismic image resolution.

In this study we use observations from surface geology (field work, Google Earth) and subsurface data (recent 2D seismic surveys and well data) to constrain the structural style and kinematics of the central Kirthar Fold Belt. We use seismic interpretation, section analysis techniques and kinematical forward modelling to constrain the balanced cross sections through

the area to show that the central Kirthar Fold Belt is driven by thick-skinned inversion which is linked with thin-skinned deformation further toward the foreland. The thin-skinned deformation pattern is dominated by folding with no major thrusts cutting through these structures. The key controlling parameters for the deformation in this area are the pre-existing structures and the mechanical stratigraphy, which is itself a result of the paleo-evolution.

## 2. Regional setting

### 2.1 Structural setting

The wider Kirthar Range area is situated on the lateral plate boundary of India. The India-Pakistan plate is moving in a northward direction respectively to Eurasia (Mohadjer et al., 2010, Fig. 1), placing the margin in an overall setting of left-



lateral transpression. The Chaman Fault, a large scale strike-slip fault, is considered to represent the lithospheric plate boundary (transform fault) in this lateral collision zone (Lawrence et al., 1981; Bannert et al., 1992). East of the plate boundary, a 150-200 km wide deformation zone is present (Bannert et al., 1992; Szeliga et al., 2009; Fig. 1). Strain partitioning is ongoing (cf. Szeliga et al., 2009 and references therein), documented by the presence of strike-slip faults as well as folds and thrusts (Fig.

1). This deformation is also reflected in the distribution of seismic active faults which show strike-slip deformation mainly towards the hinterland and dip-slip reverse faults close to the deformation front (Fig. 2, cf. Reynolds et al., 2015).

Most of the publications on the structural style of the Sulaiman and Kirthar ranges describe these to be thin-skinned fold-thrust belts. Initially, a passive roof duplex style of deformation was attributed to explain the deformation of these fold belts (Banks and Warburton, 1986; Jadoon, 1992). The associated percentage of shortening accommodated by such a thrust and duplex

dominated deformation was valued to be in the order of 37%, estimated for a section through the Sulaiman Lobe (Jadoon et al., 1992). These authors propose the same structural style for the northern Kirthar Belt. For the frontal part of the southern Kirthar Fold Belt, thin-skinned fold-thrust styles have been constructed based on field work (Schelling, 1999, see approx. location in Fig. 1). In contrast, Smewing et al. (2002a) infer, based on field work, that the southern Kirthar Fold Belt is dominated by inversion of Jurassic normal faults. Also based on field work and seismic interpretations, Fowler et al. (2004)

model the southern Kirthar Fold Belt as inversion deformation. Their sections show shortening on detachment horizons in the sedimentary cover, but the kinematic link between the shortening in the basement and in the cover remains conceptual and partly unclear.

## 2.2 Tectonostratigraphic evolution

The stratigraphic section in the study area spans from the Triassic to recent (Fig. 3), however older sediments are known along

the western margin of the Indian plate. During the Late Precambrian and Early Paleozoic, the Indo-Pakistan plate was part of Gondwana, a situation that persisted until the onset of Triassic to Jurassic rifting (Smith, 2012; Scotese, 2016; cf. Jurassic time step in Fig. 4a). Northern Pakistan was positioned at the northern margin of Gondwana facing the Phantalassic/Paleo-Tethyan Ocean. Salt deposits formed along the Gondwana margin in sub-basins, which are present and observable in the Zagros Hormuz Salt and the Salt Range Formation of Northern Pakistan (Kadri, 1995; Smith 2012). It remains unknown, if deposits (with or

without salt) from this period are present in the subsurface of the study area. Further north in Pakistan, Cambrian sediments are overlain unconformably by the Permian strata and it is not certain whether or not the intervening systems were deposited and later eroded (Kadri, 1995). During the Late Permian, the Paleo-Tethys was at its widest, indicating ongoing drifting in the ocean. The passive margin of northern Pakistan was tectonically quiescent with shallow marine to paralic conditions prevailing up to the Late Triassic (Kadri, 1995). The break-up of Gondwana which formed the Indian as well as the Afghanistan-Arabia-

Africa plates developed in the Triassic and Jurassic times (see Jurassic time step, Fig. 4a), however, the exact timing for rifting in vicinity of the study area seems uncertain.

Kadri (1995) reports that from the middle Triassic onwards the sedimentation on both sides of the Paleo Tethys suture is different. In general the Triassic mixed successions of shallow marine clastics and carbonates are grouped in the Alozai Group





or Wulgai Formation (Kadri, 1995). Continued rifting is interpreted for the Lower Jurassic deep water Shirinab Formation (which can be separated into three members Springwar, Loralai and Anjira, cf. Fig. 3). Smewing et al. 2002a find evidence for early Jurassic normal faulting due to synkinematical debris flows and slumping in the Lower Jurassic Shirinab Formation. Smewing et al. (2002b) placed the rift in the Kirthar area into Early to Late Jurassic time (Fig. 4a), mainly marked by the

successive drowning of the Springwar sandstones and mudstone cycles followed by the pelagic Loralai and Anjira members, as well as the limestones of the Chiltan Formation. The pelagic Anjira limestones are replaced eastward with the thickly bedded limestones of the Chiltan Formation. East of the Kirthar Escarpment, the pelagic Anjira limestones are not known from wells in the study area. We interpret the deep water-shallow water relationship of the Anjira-Chiltan limestones as expression of a hinge zone, related to differential post-rift subsidence. The unconformity on top of the Chiltan limestones is interpreted as

break-up unconformity (Smewing et al., 2002b). Wandrey et al. (2004) consider Jurassic or earlier extensional tectonics and failed rifting along the Indus River to contribute to buried horst-and-graben structures and the division of the greater Indus Basin into three sub-basins. The top of the Jurassic strata is marked by a basin wide unconformity (Wandrey et al., 2004; Smewing et al., 2002b). The slightly contradicting reported times for Triassic/Jurassic rifting are likely the result of several pulses of rifting related to the break-up of Gondwana.

The Cretaceous sediments are interpreted to be deposited on a westward sloping Indian shield (Kadri, 1995) in the drift phase (cf. Fig. 4b). Large deltas prograded from the emergent Indian continent, depositing the shaly to sandy Sembar and Goru formations in the middle Indus Basin (MIB) and shedding turbidites into the Kirthar Fold Belt area (Fowler et al. 2004). Portions of the Indian shield were uplifted during the Cretaceous which is partly related to the plate passing over an active mantle hot-spot (Eschard et al., 2004) which generated unconformities towards the interior of the continent. The Cretaceous

strata thin strongly towards the Jakobad High which is an intrabasinal high in the Indus Basin (Kadri, 1995) northeast of our study area. The internal structuration of the Indus Basin is interpreted as relicts of a failed rift (Zaigham and Mallick, 2000; Wandrey et al., 2004). Mixed clastic and carbonate deposits represent the Upper Cretaceous succession, consisting of the Parh, Mughal Kot and Pab Formations (Fig. 3). Island arc collision and ophiolite obduction occurred on the northwestern margin of the Indian plate during the Paleocene (Khan et al., 2009). The Muslim Bagh and Bela Ophiolites were obducted onto the Indian

margin (an island arc is anticipated northwest of the drifting Indian plate in Fig. 4b). Obduction of these ophiolites onto the Indian continental margin in western Pakistan is stratigraphically constrained between the Late Maastrichtian and Early Eocene (between ca. 67 Ma and ca. 50 Ma; Khan et al., 2009 and references herein). Likely as a result of this obduction, the shelf basin deepened and received more clastic influx. Local inversion movements are considered to be responsible for the presence of erosional unconformities. Subsequent quiescent phases are represented by widespread carbonate depositions (e.g. Dungan

Formation, Fig. 3). The northwest corner of the Indian plate started to collide with Eurasia during the Eocene (Fig. 4 C). The remaining segment of the Tethys Ocean narrowed further and eventually completely closed. In the study area this phase is represented by, on one side, carbonate deposits on the shelf edge (Laki Formation, Sui Main Limestone Fig. 3) and on the other, a westward rapidly deepening basin sourced partly with shales and siliciclastic deposits shed from the N/NW (Ghazij Formation; Wandrey, 2004; Ahmad et al. 2012). A final and short switch back to slightly more quiet conditions is indicated

by the deposition of the Kirthar Formation (limestones with intercalated shales). During the Oligocene – Lower/Middle Miocene, the Indian Ocean coastline gradually migrated southward in the foreland basin and marine conditions were progressively replaced by continental conditions (Fig. 4D). Marine conditions prevailed in the study area until the Early/Middle Miocene times and are represented by shallow marine carbonates, clastics and shales of the Oligocene Nari Formation and the

Mid-Miocene Gaj Formation (Fig. 3). In the late Miocene to Pliocene the collision between India and Eurasia resulted in the uplift of the main Himalaya and enormous quantities of clastic material reached the Lower Indus Basin (i.e. Siwaliks Group, Fig. 3). During this time, transpressive deformation along the western plate margin propagates onto the Lower Indus Basin. Recent ongoing deformation in the study area and regional uplift leads to erosion rather than deposition. Sediments along the Indus are bypassing the foreland into the Indian Ocean.

**2.3 Mechanical stratigraphy**

The behaviour of the sedimentary column when deformed is defined by its mechanical stratigraphy, which itself is the result of the tectonostratigraphic evolution. The formations deposited on the shelf margin during the drift phase are located in our study area. The presence of a long lived hinge zone in the study area results in an E-W proximal/distal sedimentological relationship of having successively more incompetent layers present towards the West.  Several detachment horizons can be

interpreted in the stratigraphic succession (Fig. 3, right column). A colour coding is given to highlight the rationale for interpreting a level as detachment.

**3. Remote field work and field work**

The core study area is covering the area east of the Kirthar Escarpment, where seismic data is available. In the east of the Kirthar Escarpment only Cenozoic rocks are outcropping and were partly investigated by field work. The area west of the

Kirthar Escarpment, where older rocks crop out, was not accessible due to security reasons. Therefore, the western area was investigated by remote methods: observations in Google Earth and remote assessment of bedding attitude data.

**3.1 Remote Fieldwork using Google Earth**

Google Earth was utilized in order to investigate structures in the study area on a broad and detailed scale. A quality check of the data quality from Google Earth revealed that images from 2010-2014 fit best to the Digital Elevation Model (DEM) (least

draping effects and offset from the DEM).

Fig. 6 shows several examples from the virtual field work with a few important observations on the deformation and mechanical stratigraphy for the study area. The strong mechanical contrast between the Eocene Kirthar limestones and the Eocene Ghazij shales is demonstrated by young gravity book-shelf faulting along an escarpment (Fig. 6a). This young, ongoing deformation is a gravitationally triggered mass movement that is a result of the competency contrast and rapid erosion of the

soft shales. Similarly, but a bit more challenging to observe, is the gravity sliding on the large anticline in the northern part of



the study area (Fig. 6b). There, slabs of Kirthar limestones (forming part of the roof of the anticline) slide downwards/eastwards across the already eroded forelimb. As a consequence, sub-horizontal Kirthar limestone beds are superposed over the steep to partly overturned younger beds of the forelimb. Further to the South (towards the background in Fig. 6b) the Kirthar beds are representing the hinge zone of the anticline. There, extensional faults are visible, including relay ramps and other associated

features which also demonstrate the young gliding motion on top of the soft Ghazij shales towards the East onto the eroded forelimb.

One example of the deformation style in the west of the Kirthar Escarpment is show in Fig. 6c. Jurassic limestones and younger strata are partly folded on different wavelengths (disharmonic folding). The large dark, gentle anticline consists of Jurassic basinal limestones. Bright limestones on the ridges of the higher frequency folds are from the Cretaceous Parh Formation. The

required decoupling and ductile deformation in-between those layers is located in the Goru Formation, known to consist of soft shales.

Consequently the observations indicate potential decollement horizons in the Ghazij and Goru Formations, as indicated in Fig. 3.

## 3.2 Remote fieldwork to assess bedding attitude data

In addition to the observations done in Google Earth, we used the "Three-Point method" to obtain additional measurements of bedding dip and strike. We used a high resolution DEM and draped satellite images to calculate bedding dip and dip-direction data from three digitized points that are located on a considered geological plane. The quality and level of detail that can be achieved is highly dependent on the quality and resolution of the input data and the outcrop conditions. We used a 30m horizontal resolution SRTM DEM (Jarvis et al., 2008) and Landsat 7 images (Landsat-7 image courtesy of the U.S. Geological

Survey). Measurements were only created in areas with univocal identification of large scale bedding planes. QC of generated data and comparison to locally existing field measurements show that bedding strike is general very reliable, whereas the bedding dip results may partly be underestimated.

## 3.3 Field work

The field work focused on collecting bedding dip data to supplement existing data and to partly QC the remotely assessed data.

Furthermore, the style of deformation as observable in field scale was investigated. Sub-recent sediments (Pleistocene to present) are tilted (Fig. 7a), being part of the large scale fold/flexure in the southern sector (Fig. 5, point a). Changes in dip are primarily apparent due to the outcrop conditions where recent outcrop degeneration was present, but small alterations of the dominant dip of the limbs of the large scale structure have been observed, possibly a result of internal thickening within the formation caused by space problems in the inner part of the folds. The necessary flexural slip has been documented even in

the very young, likely Pleistocene sediments. Fig. 7b shows striations on bedding planes in sandy beds of a tilted sand-conglomerate succession (location Fig. 5, point b). It is only in rare cases that small scale (e.g. Fig. 7c) and medium scale (e.g.



Fig. 7d) folds form (locations in Fig. 5, point c and d, respectively). These folds are interpreted to reflect higher order folds caused by local accommodation of space problems in relation to the large scale folding.

## 4. Seismic interpretation and analysis

The frontal most anticline (Fig. 5) hosts several gas condensate fields and is partly covered by 2D seismic and at least one 3D seismic cube. From 2014-2017 two new 2D seismic surveys were acquired west of this frontal anticline. For confidentiality reasons we are unable to show exact locations of the seismic lines and the well data. However, we subdivided the area into a northern and a southern sector (Fig. 5) and use two representative W-E composed seismic sections to discuss the structural differences of these sectors (Fig. 8). The seismic surveys have up to 6 km horizontal spread and up to 240 fold and utilized dynamite as the source. Processing of the lines in Fig. 8 is up to Pre-Stack Depth Migration (PSDM). These seismic lines have been tied to the vintage seismic data and wells for stratigraphic control.

The seismic quality tends to degrade towards the West and also with depth. Consequently the structural architecture in those parts is less constrained. On both sections, the top of the Eocene Kirthar limestones (cf. Fig. 3) is indicated on locations where well or seismic data unambiguously allow for that interpretation or it is constrained by surface geology (Fig. 8, orange interpretation). In the following section, a brief description of the main structures at Kirthar level is given and structures in areas of good seismic image quality are analysed.

### 4.1 Northern sector

In the northern sector, the undeformed foreland is marked by relatively horizontal reflectors (Fig. 8a, point a). Sub-horizontal seismic reflectors indicate the presence of sedimentary rocks to at least 8km in depth. A minor anticlinal feature (Fig. 8a, point b) and a more pronounced anticline (Fig. 8a, point d) are separated by a zone of discontinuous reflectors (Fig. 8a, point c), that is interpreted as a fault offsetting the Kirthar limestone. The relative timing of these frontal structures is well depicted in the growth strata imaged on a time-domain seismic section nearby (Fig. 9):

The interpreted growth strata packages GS1 and GS2 show westward thinning and onlap as well as progressive limb rotation related to the triangle deformation at depth (with thrust "1" and roof thrust "2", Fig. 9). The youngest growth strata package, GS3, is only deformed above the projection of fault "3". As a result of the location of this deformation, this fault is younger than faults "1" and "2". Movement on this fault generates a fault-propagation fold in the hanging wall. The good seismic image allows to define the stratigraphic level of the roof thrust (fault "2" in Fig. 9). The thrust has a trend that is parallel to the bedding in the Paleocene shales (upper Ranikot shales, cf. Fig. 3) just below the thick and competent limestones (Sui Main limestones, cf. Fig. 3), characterized by the low-reflectivity seismic character. Tilting of the reflectors below thrust "1" is partly due to a velocity pull-up, but some additional tilting related to layer parallel shortening in deeper levels cannot be excluded. The deeper parts of thrusts "1" and "3" is relatively uncertain based on the seismic profiles. However, the Jurassic Chiltan Formation is drilled in the hanging wall of thrust "3", indicating that the thrust cuts below the Jurassic.



A syncline is located west of these frontal structures, though it is not imaged on the seismic due to steeply dipping to overturned beds (Fig. 8a, point e). A large scale anticline with Kirthar limestones on surface level is indicated at point f (Fig. 8a, cf. Fig. 5). The low-reflectivity seismic facies below the Kirthar limestones are Eocene Ghazij Shales (Fig. 8a, point g). These shales thicken dramatically from wells in the East (several tens of meters) towards the West (several hundreds of meters, constrained

by outcrop and seismic velocity data). A small scale anticline of higher order is cropping out with Kirthar limestones on the surface (Fig. 8a, point h). A syncline is marking the western end of the seismic (Fig. 8a point i). Kirthar Formation is cropping out also at the Kirthar Escarpment west of the section (Fig. 5 northern sector, at approx. 1150 m above sea level). It is notable that the regional elevation of the Kirthar limestone increases from east to west (approx. 2.5 km in the western syncline, Fig. 8a point i). At the highest outcropping point of Kirthar limestones the difference to the estimated regional level is 5.5 – 6 km

with an uncertainty related to the interpreted slope of the regional elevation. A rough depth to detachment analysis conducted with the excess area approach (Epard and Groshong, 1993) on the large scale anticline reveals an upper detachment depth of 8-10 km. The spread in the predicted detachment is due to high uncertainty in the deeper stratigraphic picks on the seismic and the fact that the Kirthar Formation is not returning to regional elevation in the syncline to the West.

**4.2 Southern sector**

In the southern section, the undeformed foreland (Fig. 8b, point a) shows sub-horizontal reflectivity to at least 8 km in depth. To the West a minor flexure (Fig. 8b, point b) is situated underneath a seismic noise zone hiding a thrust fault (Fig. 8b, point c). This anticline (Fig. 8b, point d) is the southern along-strike continuation of the anticline on the northern section (Fig. 8a, point d). The syncline towards the West (Fig. 8b, point e) is much broader than its northern equivalent. Sub-horizontal reflectors indicate the presence of sedimentary rocks to at least a depth of 10 km. The frontal structures have been analysed and

interpreted. Finally the concluded model is illustrated and tested by running a kinematic forward model (Fig. 10). The interpreted fault geometries as well as a stratigraphic template elaborated from wells and outcrop sections are used for the starting configuration of the model (Fig. 10a). Step 1 follows the sequence elaborated in the northern sector (cf. Fig. 9)which shows a small fault-bend fold that forms a small triangle structure at the deformation front (Fig. 10b). This triangle structure is cut by a subsequent thrust, forming a fault-propagation fold (figure 10c). This step is modelled using the tri-shear

implementation in Move software (Midland Valley, 2016, Fig. 10c). There are several parameters that control the shape of the anticline. In detail, more than one solution (combination of parameters) can generate an approximate fit to the given constraints (seismic, well data (not shown) and surface dips), but differences are not significant. The reasonable fit shown in Fig. 10c supports the fault interpretation and the amount of shortening applied to these frontal structures (about 5000 m of horizontal shortening). To the west of the red stippled line in Fig. 10c, the model does not exactly match the seismic - the interpreted

Kirthar Formation is constantly rising until the limestones crop out in a small scale fold (Fig. 8b, point f). West of point f the Kirthar formation continues outcropping to the Kirthar Escarpment (just west of the end of the seismic line) at an elevation of around 1850 m above sea level.



Similar to the northern section is the westward thickening of the Eocene Ghazij shales. The shales are thin (several tens of meters) in the frontal anticline (Fig. 8b, below point d) and thicken towards the West (Fig. 8b, point g). This thickening is taking place especially west of the clinoforms (Fig. 8b, point h), which mark the carbonate margin of the Sui/Laki Limestones. On the southern section the regional elevation to the West rises constantly from the syncline axis (Fig. 8b, point e) towards the

Kirthar Escarpment. The structural elevation gain above regional at the Kirthar Escarpment is more than 6500 m (with uncertainty related to the interpreted regional level, Fig. 8b stippled orange line).

## 5. Linking thick-skinned and thin-skinned deformation

Despite some structural differences between the sectors, the common observation is the overall increase of the regional elevation level of the Kirthar and other formations from the East to the West. Such an increase in elevation can potentially be

explained by several mechanisms: a) a strong wedge shape of the pre-deformational strata below the Kirthar, b) a thrust/fault to a deeper structural level, c) internal structural thickening of formations below the Kirthar Formation or any combination thereof as these proposed mechanisms are not mutually exclusive.

We suggest that the order of structural uplift (larger than 5500 m) is linked to a deeper structural level in the basement. The other mechanisms above might still contribute a smaller part to the regional uplift. This anticipated thick-skinned deformation

needs to be linked to the demonstrated thin-skinned deformation close to the deformation front.

In order to generate a most reasonable structural model we briefly review the available nodal planes of earthquakes as well as some indications from the geological map west of the Kirthar Escarpment.

### 5.1 Focal mechanism from the southern western fold belt

We use ISC bulletin derived nodal planes to constrain potential fault geometries in the subsurface in the wider study area (Fig.

2). Given the tectonic setting of a lateral collision zone, it is not surprising that earthquakes towards the current plate boundary at the Chaman Fault document dominantly strike-slip faulting. Some focal mechanisms of earthquakes close to the deformation front show dominant dip-slip shortening (Fig. 2 and Table 1). It is not clear which of the two nodal planes was the moving plane. We could either assume it is the one with the lowest dip or it is westward dipping corresponding to the eastward directed shortening. For the first assumption fault dips are between 15° - 45° and for eastward dipping faults between 15° - 57°. In both

cases, the steeper faults are considered to be too steep to represent newly initiated reverse/thrust faults. We interpret these steep faults therefore as reactivated pre-existing faults.

### 5.2 Deformation pattern west of the Kirthar Escarpment

The outcropping structures west of the seismic coverage/west of the Kirthar Escarpment yield some indications about the structural architecture below the Kirthar and Ghazij Formations (Fig. 5). The area west of the Kirthar Escarpment has a high

mean elevation (more than 1000m above sea level). The anticlines with Jurassic outcrops represent the structures with the





highest elevations in the area (labelled with bold numbers 1 in Fig. 5). In-between are areas where Paleocene (and sparsely also Eocene) rocks are preserved, which represent relative structural lows (labelled with bold numbers 2 in Fig. 5). A further characteristic is the presence of long wavelength folds with several km wavelength (labelled with bold numbers 3 in Fig. 5) and anticlines with much smaller wavelengths and higher frequencies (labelled with bold numbers 4 in Fig. 5), indicating a

much shallower detachment horizon. The large scale anticlines are usually double plunging and have roughly N-S trending axes, but a variety of additional directions are present as well. A plausible deformation model should be able to explain this complex pattern.

### 5.3 A simplified thick-skinned - thin-skinned inversion model

We propose that an inversion model is the best solution to explain all the different observations and constraints. Yamada and

McClay (2004) demonstrated that the shape of the normal fault with its syn-kinematic fill (Fig. 11a) defines the shape of the inversion anticline (Fig. 11b). Interesting to note is the presence of double plunging anticlines and the possibility of local lows in-between the anticlines and towards the hinterland (Fig. 11b), which can be compared to the structural pattern west of the Kirthar Escarpment in map view. The analogue experiments are limited by the rigid and non-deformable footwall whereas in nature, this is likely not the case. In the study area the shortening of the inverting normal fault is considered to be transferred

to a detachment in the sediments (Fig. 11c), explaining the presence of short wavelength folds adjacent to the large wavelength folds (cf. Fig. 5). We further suggest that the complex map pattern is likely the result of a much more complex inverted fault pattern as observed in natural rifts. En-echelon pattern and overlaps of faults with intact and broken relay ramps, horses etc. (Fig. 11d) could contribute to a more complex deformation pattern if inverted. Additionally, several stacked detachment horizons allow to accommodate shortening without the necessity of major thrust faults breaking the surface. We consider that

the inverting faults are inherited from the original rift phase on the lateral boundary when the Indian plate rifted from northern Godwana (Fig. 4a). The direction of the rift faults thus would also define the N-S direction of the anticlines, which is strongly oblique to the plate kinematic vector.

### 5.4 Southern section kinematic model and balanced section

The kinematical model of the frontal deformation structures in the southern section (Fig. 10) accommodates approximately

5000 m of shortening. However, this amount of shortening is not enough to explain the 6500 m of regional uplift towards the hinterland at the Kirthar Escarpment when taking into consideration reasonable fault dips. A fault with a 45° angle and a displacement of 5000 m would generate a structural uplift of 5000 m from a simple geometric perspective. Either a much steeper fault is necessary, or some additional shortening above/in front the inverting fault are required to explain the 6500 m of regional uplift. From careful seismic interpretation and dip-analysis we have interpreted the presence of small passive roof

duplexes underneath the soft Eocene Ghazij shales. Below point g in Fig. 8b some strong but laterally discontinuous reflector packages are present (between points h and i). The reflectors are interpreted to represent the Paleocene limestones. The discontinuous pattern is interpreted to be caused by poor imaging and by structural imbrication. East of point i (Fig. 8b), a





small back-thrust is interpreted. The structural solution is presented in the balanced section (Fig. 12a). By adding the shortening of these small passive roof duplexes to the total displacement on the basement fault, the required regional uplift at the Kirthar Escarpment can be achieved, as is demonstrated in Fig. 13.

The simplified kinematical model shows the evolution starting from one major normal fault (Fig. 13a). For simplicity we will assume syn-kinematical growth in the lower Jurassic formations, although the fault might have been active as normal fault earlier and later as well (see Tectonostratigraphic evolution section).

The frontal triangle and the small identified back-thrust are likely formed in an early deformation phase. In the model, this deformation is linked to slight inversion of the displayed normal fault (Fig. 13b), however, the deformation could also be linked to shortening further in the West which is transferred via thin-skinned detachments. The main inversion of the normal fault generates shortcut faults with a slightly smaller dip angle in the sediments of the footwall. The presence of several weak stratigraphic units allows some wedging as well as the passive roof backthrust in the Ghazij shales (Fig. 13b and c). This stage reflects large scale layer parallel shortening of the stratigraphy above/in front of the inverting normal fault. With increasing inversion above the null point, the pressure on the basement in the footwall likely increases and it finally yields. A basement shortcut develops and links with a suitable detachment therefore generating the frontal anticline (Fig. 13d). The features/constraints visible in the seismic and at surface, and especially, the regional elevation uplift at the Kirthar Escarpment can all be explained with one major fault inversion (Fig. 13 D). West of the Kirthar Escarpment the regional elevation remains relatively high and is not dropping as in the simplified model (Fig. 13d hatched area). For that area, additional shortening associated to inverting normal faults is required.

The balanced section in Fig. 12a shows the final interpretation for the seismic displayed in Fig. 8b, with further details than shown in the kinematic section (Fig. 13). This section restores to the pre-contractional situation as shown in Fig. 12b. The balanced section has been constructed by line length restoration onto a carefully constructed stratigraphic template that takes well data and regional thickness trends into account. An overall areal balance has been considered. The amount of shortening is about 10 km, corresponding to 20% between the fixed and the loose line. The Eocene and Oligocene strata has shorter line length than the older strata due to roof back-thrusting on the Eocene Ghazij shales and subsequent erosion. The loose line in Fig. 12b is not absolutely straight for Paleocene and older, documenting a small remaining error in the order of 1% which is within the drafting accuracy. The upper part of the section can be considered as well constrained (seismic image, surface geology and well control). The deeper part of the section is constrained by kinematic and balancing considerations. In detail, there could be other solutions fulfilling the constraints (e.g. a more complex pattern of imbrications/duplexes and other structures accommodating layer parallel shortening). In that sense, the presented section is a likely scenario honouring as many constraints as possible and remaining as simple as possible.

### 5.5 Northern section kinematic model and balanced section

As demonstrated above, the northern sector is dominated by folding and shows a more gradual rise of the regional elevation towards the West. We interpret the shape of the large anticline in Fig. 8a as an uplifted detachment fold (Fig. 14) with plastic




deformation in the fold core. In detail the deformation in the fold core can be accommodated by small scale thrusts (e.g. fishtail wedges etc.). The relative gentler uplift of regional elevation towards the West indicate the presence of several small inverting faults in the basement in comparison to the southern section. A proposed kinematical evolution is displayed in Fig. 15. Several half graben normal faults with thickening strata toward the fault (again in Jurassic for simplicity reasons) are present below

some post rift strata (Fig. 15a). The thick-skinned movement is linked with a thin-skinned decollement close to the base of the sedimentary column (Fig. 15b). For simplicity, this step is modelled with one footwall-shortcut on the westernmost fault. The large scale anticline in the northern sector likely starts to grow in this increment. Fig. 14b anticipates a buttressing effect of the easternmost fault causing the folding. Alternatively, early inversion movement on this easternmost fault could generate a perturbation in the sedimentary sequence which is subsequently amplified by shortening which is transferred along the basal

décollement. The inversion is modelled with a foreland propagation sequence. The second (middle) fault is also modelled with a small footwall shortcut (Fig. 15c). The associated shortening is amplifying the large scale fold. Some shortening of this increment might cause the observed triangle deformation (Fig. 9). Finally, the easternmost fault is inverting including a footwall shortcut that links to the observed frontal structures (Fig. 15d). The final balanced section honours some more details as constrained by the seismic (Fig. 14a). In total, the northern section is less constrained than the southern section as the seismic

is allowing more solutions for the deeper geometry. The main uncertainties are the amount of basement faults, the amount of initial extensional throw, how many faults and shortcuts are present in the sediments and the sequence of deformation propagation. The kinematic model in Fig. 14 as well as the balanced section are thus not unique solutions however they provide a satisfactory explanation for the observed structures that is consistent with the mechanical stratigraphy, the regional observations and the local constraints (seismic and surface geology).

The balanced section of the northern sector (Fig. 14a) restores to the pre-contraction geometry as is shown in Fig. 14b. The method of restoration is the same as for the southern sector. The weak formations have been additionally areally balanced. The amount of shortening is approximately 11.2 km, corresponding to 18% between the fixed and the loose line. The loose line in Fig. 14b is not absolutely straight, documenting a small remaining error in the order of the drafting accuracy.

## 6. Discussion

In the following section we discuss implications of the results from the local to regional scale, compare the deformation style to similar fold belts, and finally address some uncertainty issues.

### 6.1 Local paleogeographic controls on deformation

On a local scale the deformation seen in the geological map (Fig. 5) is partly mimicking the original rift geometry. The large scale folds are most likely representing the former location of the main extensional growth grabens. The dominating N-S

orientation of fold trends is thus is directly controlled by the former rift geometry and thus has a strong influence on how strain is partitioned on this lateral margin.





In detail, most local differences in the structural style and orientation seem to be based on slight paleogeographic differences as well. Comparing the northern to the southern sectors of the study area, there is difference of the detachment depth of the trailing thin-skinned deformation. In the South, the large syncline (Fig. 8b) indicates a flat segment of the detachment in lower Jurassic or Triassic rocks. The comparative thrust fault clearly cuts deeper in the North (Figs. 8a, 9, 14). As a consequence the

frontal anticline shows a south to north along-strike structural uplift. The uplift has been recognized for a long time (wells and the distribution of the existing gas condensate fields), but geometrically they have not been properly investigated. The reason for the along strike change of the basal detachment, however, is not known. It may have geometrical (fault throw, angle, depth) or facies (mechanical stratigraphy) related reasons. Both, however would be inherited from the pre-contractional evolution, with the rifting phase likely having the greatest impact.

There is one very clear example on how the long lived hinge zone and the associated facies changes control the young contractional deformation. The tip of the triangle/duplex in the centre of the southern section is localised at the point where the Laki Formation limestones (Sui Main Limestones, stippled orange line) have their paleo-shelf edge and are replaced laterally by Ghazij Formation shales (Fig. 12a, see also Fig. 8b, point h). The limestones have several hundred meters of thickness in the frontal anticline and are overlain by several tens of meters of Ghazij shales only. West of the Kirthar

Escarpment, the Laki Formation/Sui Main Limestones are missing, instead several hundred meters of Ghazij shales are present. The clinoforms of the Sui Main Limestones are well imaged (Fig. 8b point h). West of that point most of the Laki Formation is replaced by marls and shales that act as passive roof thrust for the inversion related footwall duplexes underneath. The juxtaposition of carbonate margins that border basinal facies can localize thrust faults as has been demonstrated by centrifuge physical modelling by Dixon, 2004.

**6.2 Kirthar Fold Belt deformation**

Our model of inversion with linked thin skinned deformation for the central Kirthar Fold Belt is in line with the observations and the model proposed by Smewing et al. (2002a) and Fowler et al. (2004) for the southern Kirthar Fold Belt. In our study we demonstrate how thick-skinned inversion and thin-skinned deformation kinematically link to produce the observed deformation pattern. With our model we are also able to explain an observation of Smewing et al. (2002a) where they describe

field evidence of a Jurassic normal fault that is still under net extension, despite the assumed inversion and relative high structural elevation. Following our model the upper part of the former normal fault could remain in net extension and be significantly uplifted above their original regional elevation due to footwall imbrication and shortcuts (cf. the former normal faults in Fig. 14). These imbrications however, do not penetrate to the surface but generate structural wedges with a roof thrust in Ghazij shales. They are unlike the passive roof duplexes proposed originally for the northern Kirthar Fold Belt by Banks

and Warburton (1986). Those authors use classical thin-skinned fold-trust belt geometries based on the sequential imbrication of the foreland sequence above a pre-Jurassic continuous planar detachment horizon. Their roof thrusts are localized in Ghazij shales and further towards the hinterland in Goru formation shales. Shortening in such a system often approaches 40-50%, a value much higher than the shortening observed in the central (our study) and southern Kirthar Fold Belt (Fowler et al., 2004).



The large scale map pattern does not significantly change from our study area towards the northern Kirthar Fold Belt. We propose, that the deformation observed in the northern Kirthar Fold Belt (i.e. Banks and Warburton, 1986) could also be caused by linked thick-skinned and thin-skinned inversion related deformation. Similarly, the thin-skinned deformation observed by Schelling (2000) can be put into this context. The sections investigated by Shelling are relatively short and thus only cover the leading edge margin of the thin-skinned deformation (similar to the frontal structures in Fig. 12). Thus, no major south to north discrepancies in shortening values need to be considered for the Kirthar Fold Belt. The deformation style does not necessarily vary dramatically, however the way the shortening is accommodated is considered to be controlled by local inherited controls.

## 6.3 Possible lithospheric inheritance of the inversion belt

It is important to briefly discuss some potential reasons why the Kirthar Fold Belt is dominated by inversion with thin-skinned deformation instead of following classical thin-skinned fold-thrust belt model. The importance of a structural inheritance from rifting has already been proposed by Smewing et al. (2002a). However, most pro-wedge thrust belts affect areas which went through rifting and passive margin settings before collision. Whether the continental margin in the collision phase evolves into a dominated system of thin-skinned or thick-skinned deformation depends on several factors.

The presence of a weak (ductile) middle or lower crust seems to be key factor which allows for distributed deformation through most of the crust, which results in forming fold-thrust belts with a dominant/primary thick-skinned character (Lacombe and Bellahsen, 2016). Thermally weakened shear zones might be conserved in little extended proximal continental margins which can also influence the deformation style. Weak crustal levels are often lacking in distal parts of the margins as a result of the rheological evolution of the rifted margin over time (Perez-Gussinye and Reston, 2001; Cloetingh et al. 2005; Reston & Manatschal, 2011). The resulting stronger lithospheric domains are more prone to localized deformation in a continental subduction style (Lacombe and Bellahsen, 2016). Thus, the relative position and the time since rifting apparently play a role in determining in which mode the convergent deformation will reactivate structures.

We therefore speculate, that the inversion dominated Kirthar Fold belt represents the inner part of the continental margin in which a weak continental crust is still present. The long lived hinge line observed in several facies associations is interpreted to reflect approximately the limit of the major post-rift subsidence and the eastern border of the extended lithosphere (assuming pure-shear). The more than 100km wide area from the deformation front to the Ghazaband fault (Fig. 1) is interpreted to be dominated by initial inversion (later partly overprinted by strike-slip deformation). We infer that this large area shares a similar rheology which was inherited from Gondwana and the break up phase. The width of this zone might indicate that the lithosphere rifted in a wide rift mode (Buck, 1991) before continental break-up to the West. Consequently there should be a narrow zone of highly extended crust (external rifted margin) present west of the inversion belt. Today this zone is covered by Flysch sediments, bracketed between the Ghazaband and Chaman Faults (Bannert et al., 1992). How much of the former external rifted margin has been subducted or laterally displaced along the strike-slip faults remains difficult to estimate and is beyond the scope of this paper. Interestingly, further to the North in the Pamir area earthquake tomography data is interpreted to show delamination and rollback of the Indian plate lithosphere (Kufner et al., 2015). India's thinned western continental




margin separates from Cratonic India and subducts beneath Asia while the buoyant northwestern salient of Cratonic India bulldozes into Cratonic Asia (Kufner et al., 2015).

## 6.4 Hybrid thick- and thin-skinned systems in other areas

Thick-skinned inversion of passive margin or intra-cratonic rifts is considered to be present in 50% of orogens with documented deformation style (Nemčok et al., 2013). There are various possibilities how thick-skinned deformation can contribute to the deformation of a fold-thrust belt (see recent review of Lacombe and Bellahsen, 2016 and references therein). Here, we briefly compare the deformation style elaborated for the central Kirthar Fold Belt with other well constrained examples of linked inversion with thin-skinned deformation.

For the Malargüe Fold-Thrust belt, Giambiagi et al. (2008) revealed that the reactivation of normal faults was coeval with the activation of shallow detachments and low-angle thrusting at the thrust front with several faults moving at the same time in some portions. Also for the Malargüe Fold-Thrust belt Fuentes et al. (2016) work out geometric relationships of the hybrid system with a series of detailed sections based on surface geology, seismic and well data through that thrust belt. Their section "E" show strong similarities with the deformation style in the southern section of our study, especially the imbrication of sediments in the footwall with duplexes and a passive roof thrust on top. Recently, Mahoney et al. (2017) proposed a very similar deformation for the Eastern Muller Ranges in the Papuan Fold Belt in Papua New Guinea. There, the Cenozoic carbonates are shortened to around 13-21% but are partly uplifted up to 7km above regional elevation. Mesozoic rift faults, partly inverting and partly linking to thin skinned detachments are considered to reflect the major control on deformation. Triangle structures and back-thrusts are considered transient deformation steps, related to the uplift and erosion history while the deformation accumulates before linking to the frontal deformation structures (Mahoney et al., 2017). The proposed deformation is very similar to the style we consider for the central Kirthar Fold Belt. In our example the presence of a complex mechanical stratigraphy with several detachment horizons in the stratigraphic column seems to produce even more complex geometries than in the example from Papua New Guinea.

## 6.5 Uncertainty

Basement involvement is very often used in balanced section to account for regional elevation uplift towards the hinterland. However, for a relatively small uplift of regional elevation the thick skinned explanation is often ambiguous, as there are often several alternative possibilities which are not investigated (e.g. strong wedging of the pre-kinematic sedimentary sequence, change in basal detachment depth, change in basement dip etc.).

Recently, Butler et al. (2018) demonstrated that for several reviewed sections that there is a substantially greater range of solutions available for interpreting the geometry and evolution of thrust belt structures than implied by the original idealized models. For a specific section in the Papuan Fold and thrust belt two realisations are available by different authors. One with thin-skinned and one with inversion style tectonics (e.g. Hill, 1997; Buchanan and Warburton 1996; cf. Butler et al. 2018).



Similar there is a strong discussion on the contribution of thick-skinned deformation below the different segments of the Zagros fold belt (see discussions in Lacombe and Bellahsen, 2016; Hinsch and Bretis, 2015, for the Mountain Front Flexure).

In order to overcome limitations from single deterministic geometries, Butler et al. (2018) propose good documentation, alternative models and to embrace the uncertainties. In this work, we show the original seismic data, review in detail the

regional to local context and use these as arguments why we think our presented deformation model is the most plausible for the central Kirthar Fold Belt from the other investigated alternatives. We do not show alternative models, but we highlight our workflow, the considered constraints and indicate uncertainties of the sections. With all the arguments given, the contribution of deep founded faults with associated thin-skinned deformation can be considered as reliable, and the pure thin-skinned deformation style can be considered as obsolete. In detail the interpreted and constructed sections are as good as the constraints

allow and thus still have several solutions. The amount of uncertainty in the sections depends also on the level of observation.

## 7. Conclusions

Large scale strain partitioning along the western Indian plate leads to major left lateral strike-slip faulting close to the plate margin as well as to NW-SE to W-E shortening close to the deformation front of the Kirthar Fold Belt. We analyse regional (focal mechanisms, geological maps) to local (reflection seismic and well data, surface geology) data at the front of the central

fold belt to constrain the structural architecture and style. The deformation is controlled by the inversion of inherited rift faults, likely of Jurassic age, which is buried underneath the sediments. The young shortening on the rift faults is coupled with thin-skinned deformation by imbricating and shortcutting into the footwall and transferring some shortening onto a detachment horizon. As a consequence, large scale folds build as a result of the thick-skinned inversion and smaller scale folds and thin skinned related thrust deformation form in front. In the southern sector a structural elevation gain of approximately 6500 m

across one large monocline clearly indicates the influence of the deep seated faulting. Towards the North the structural elevation gain is distributed across several folds indirectly related to several inverting faults at depth. The main control on deformation is the presence and orientation of the pre-shortening rift. In addition, the rift and post rift history resulted in some prominent E to W proximal to distal facies trends being reflected in a heterogeneous mechanical stratigraphy which is responsible for the style on how shortening is accommodated in the thin-skinned structures.

The hybrid deformation style of thin-skinned to thick-skinned deformation is also present in other fold-thrust belts around the world with hydrocarbon resources. Combining as many constraints as possible from regional to local scale facilitates the development of plausible structural models and assess uncertainties. The importance of understanding the structural architecture and kinematics is here and there of paramount importance for the successful exploration of these resources.



## Acknowledgements

Several colleagues or former colleagues at OMV Pakistan contributed feedback, local knowledge and assistance in the course of the project, i.e. Waqas Ahmed, Muhammad Aamir Rasheed, Muhammad Ibrahim. Also colleagues at OMV in Vienna where involved in the project or contributed through discussions: e.g. Peter Hagedorn, Zsolt Schleder, Klaus Pelz, Wolfgang Thöny,

Bernhard Bretis, James M. Kiely, Noah Stevens and Maziar Haghighi. Cameron Sheya is thanked for proof reading and improvements on the language. The Mehar EL Joint Venture [OMV Pakistan/UEP, OPL, ZPCL, GHPL] is thanked for the permission and opportunity to publish the study.

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





**Figures and captions**

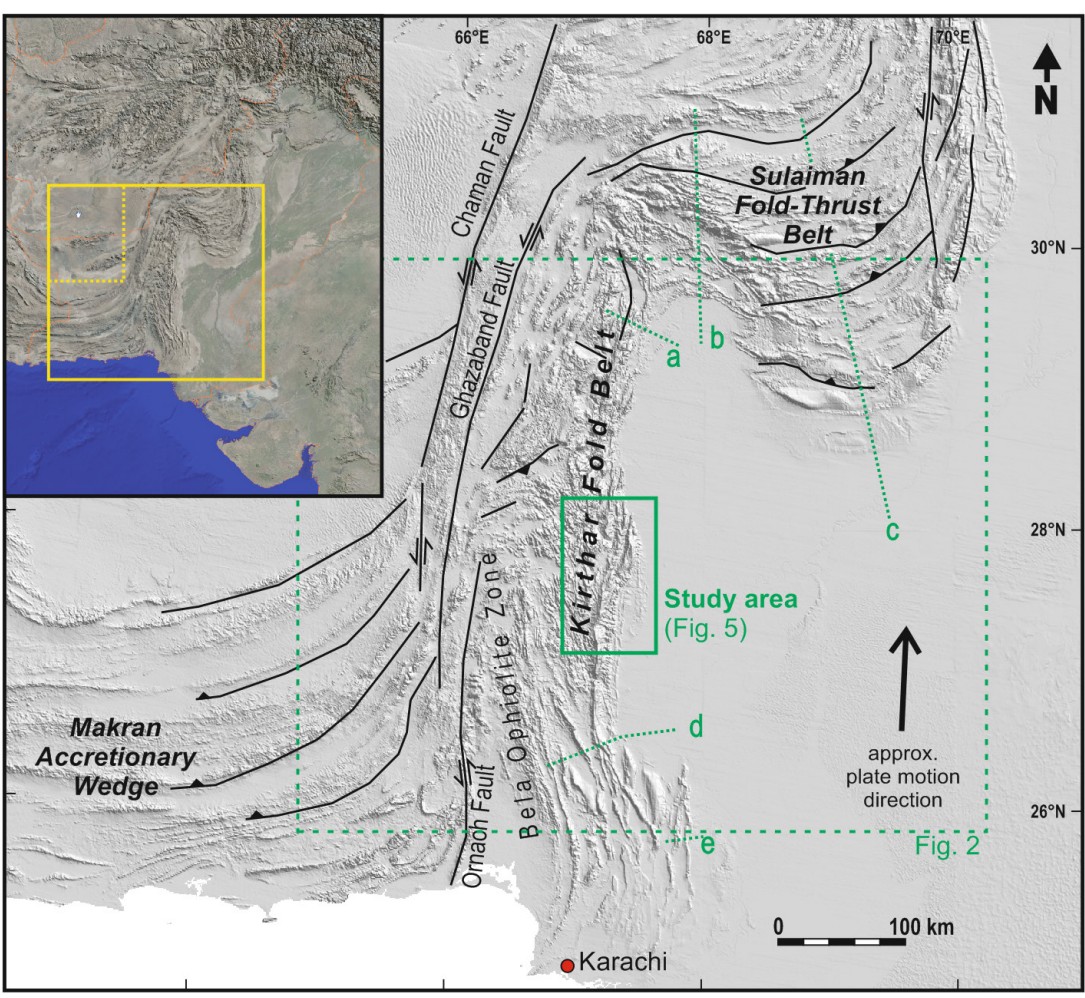

**Figure 1: Simplified structural sketch of the wider Kirthar fold-belt area on a shaded relief map. Location is indicated in in the inset map. The approximate plate motion is from Mohadjer et al. (2010); Sections from other studies are indicated: a+b: Banks and Warburton (1986), c: Jadoon et al. (1992), d: Fowler et al. (2004), e: Schelling (1999).**





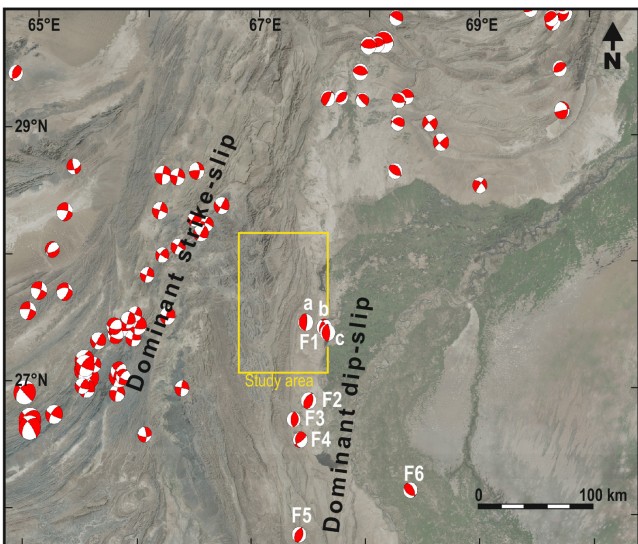

**Figure 2: Nodal planes from International Seismological Centre (2015) plotted on satellite image. Location of the figure is indicated in Fig. 1. Labelled events are listed in Table 1.**



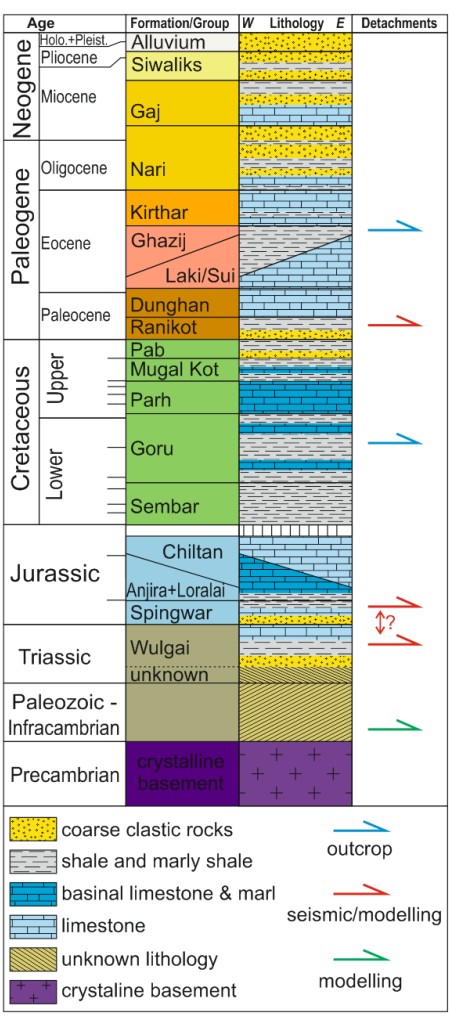

**Figure 3: Litostratigraphic overview with hydrocarbon play elements and mechanical stratigraphic interpretations (after Kadri, 1995; Tectostrat 2001; Smewing et al. 2002b and author observations).**



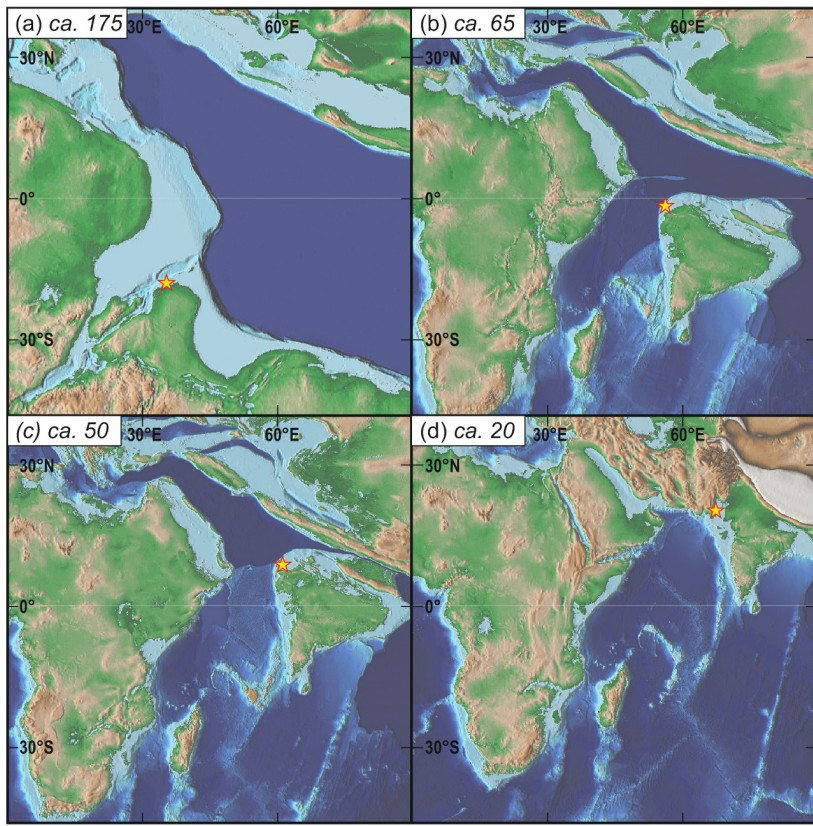

**Figure 4: Paleogeographic evolution of the study area as part of the Indian plate since the Jurassic. (a) Jurassic, ca. 175 million years, a rift evolves northwest of the approximate study area location (star; map from Scotese, 2014a), (b) ca. 65 million years, Cretaceous/Paleocene, drifting northward. An Island arc is visible north of the approx. study area location (map from Scotese, 2014b). (c) ca. 50 Eocene, post ophiolite obduction, but pre-collision with Eurasia (map from Scotese, 2014c). (d) ca. 20 Miocene, early collision stage with flexural foreland stage (map from Scotese, 2014c).**



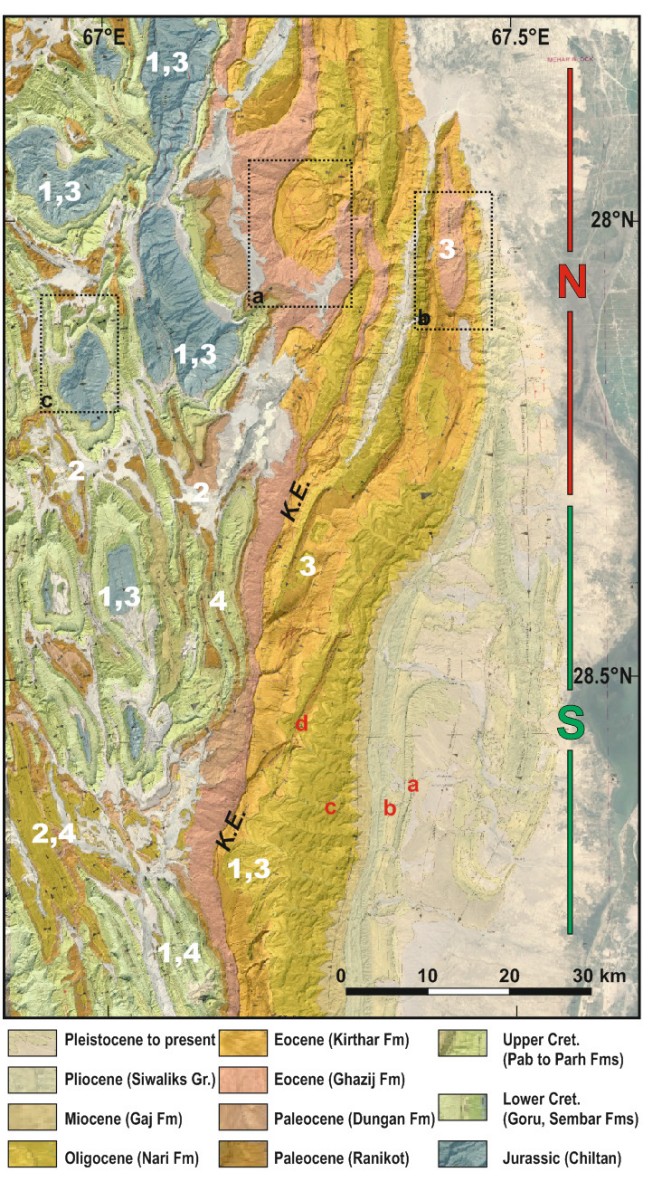

**Figure 5: Semi-transparent lithostratigraphic map (modified after Tectostrat, 2001) of the study area draped in Google Earth. Dotted rectangles with labels a, b, c indicate approx. areas seen in slanted view in Fig. 6a-c. Red labels a-d indicate locations of field photographs in Fig. 7. Northern and southern sectors of the fold belt are indicated by bold red and green lines, respectively. Bold white numbers indicate examples for 1: structural highs, 2: structural lows, 3: long wavelength anticline, 4: short wavelength folding.**



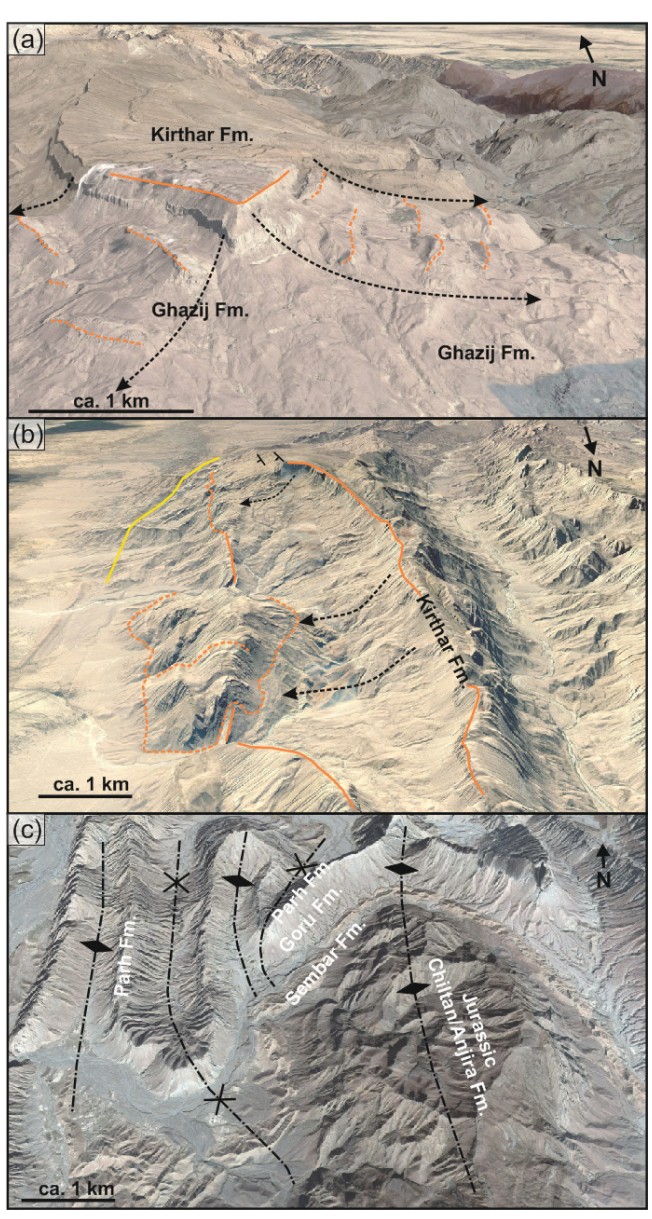

**Figure 6: Example for remote field work with Google Earth. Locations of the views are indicated in Fig. 5. (a) Recent mass wasting: Blocks of Kirthar Limestones glide down the eroded flanks of soft Ghazij Formation; (b) Sub-recent to recent mass wasting: a large slab of Kirthar Limestones from the anticline roof is now folded over the previously eroded forelimb of the anticline. In the background extensional faults are visible on the roof of the boy-fold anticline in the Kirthar limestones. The limestones partly glide over the vertical beds of the forelimb. (c) Disharmonic folding: Jurassic rocks show large wavelength folding, while the hard limestones of the Cretaceous Parh formation are folded in smaller wavelength and higher frequency. A weak decollement zone is located in the Goru Shales.**



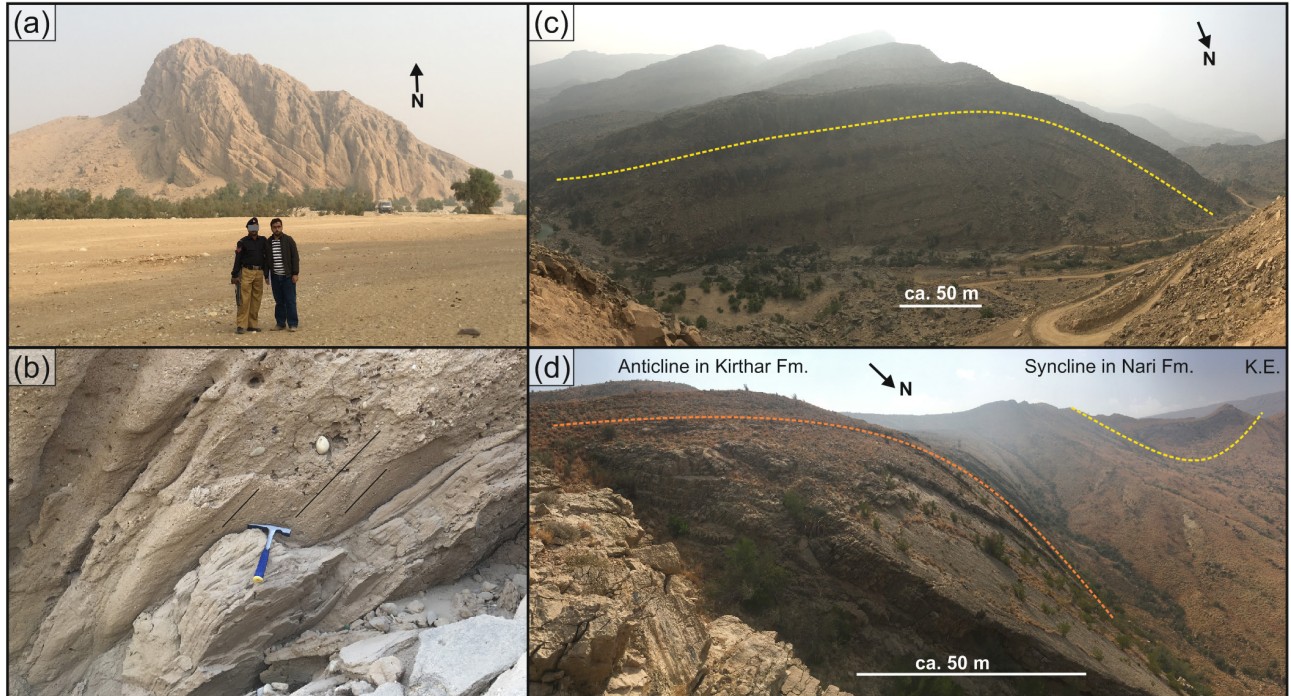

**Figure 7: Examples for observations from fieldwork. (a) Sub-recent conglomeratic sediments are folded and eroded. (b) Striations on a bedding plane in (Pleistocene?) conglomerates indicating flexural slip folding. (c) small scale anticline in Nari Formation rocks. The amplitude and wavelength of the fold suggest, that the lower detachment horizon is likely in lower Nari Formation. (d) small to medium scale folding in Kirthar and Nari Formations. The fold is a mappable feature (cf. Fig. 5) and indicates to a detachment horizon below the Kirthar limestones.**



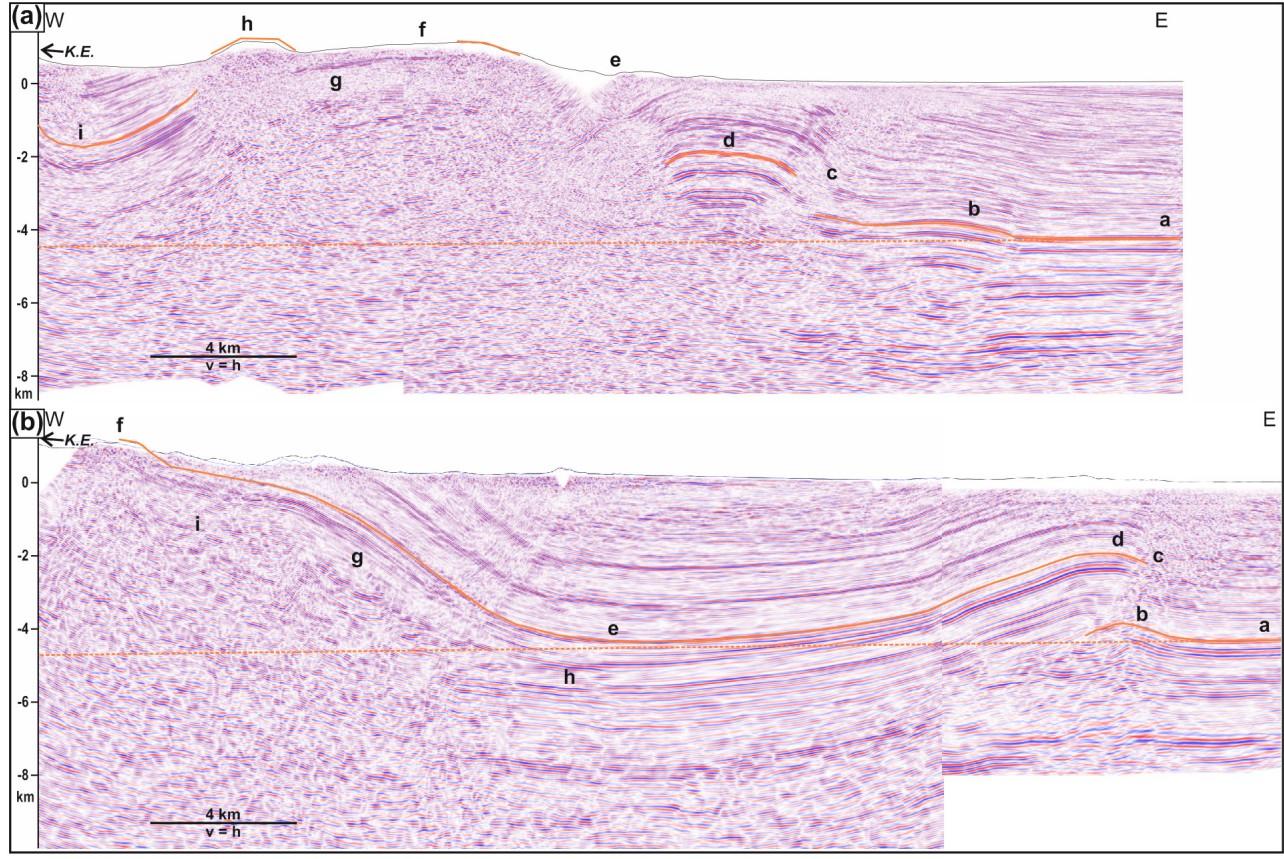

**Figure 8: Two W-E composed seismic sections of pre-stack depth migrated seismic. Orange interpretations indicate clearly constrained Top Kirthar Formation from seismic, wells and outcrop. Stippled orange line is anticipated pre-contractional regional elevation of the Kirthar Formation. K.E.: Kirthar Escarpment. (a) Seismic section composed from two overlapping 2D seismic lines in the northern Sector (exact position not shown for confidentiality reasons). Labels a-i are used to indicated features discussed in the text. (b) section composed from 2D and 3D seismic data. Labels a-i are used to indicated features discussed in the text.**



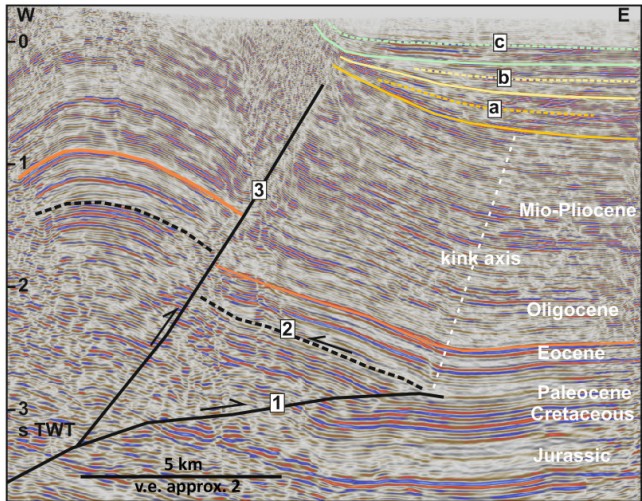

**Figure 9: 2D seismic section in time domain in the northern sector (exact position not shown for confidentiality reasons) with fault and growth strata interpretation. Faults 1-3 and growth strata packages GS1-GS3 are discussed in the text.**



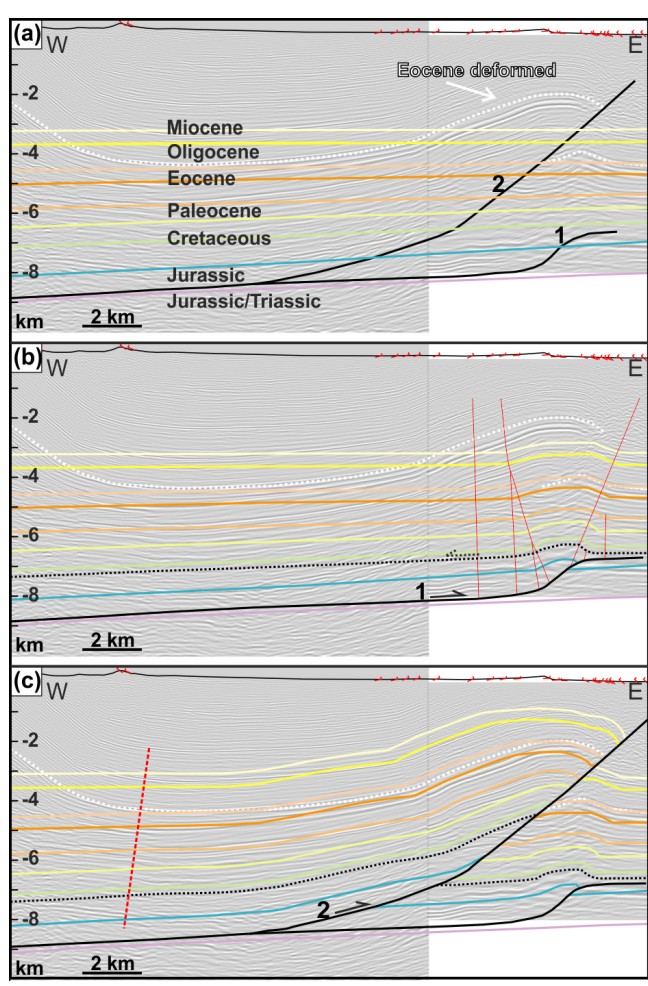

**Figure 10: Forward modelling of the frontal structures in the southern sector (a) The seismic image, surface geometry including dips, the present day deformed state of the Top Eocene limestones (constrained by nearby well control) as well as the interpreted and constructed faults are given as reference frame for the forward model. The model uses a stratigraphic wedge with thicknesses which are constrained by well and outcrop observations. (b) A small triangle structure at the deformation front is modelled with fault-bend folding. The lower detachment (1) is in lower Jurassic or Triassic succession, the upper one is interpreted in the soft Cretaceous Goru shales (dotted line). (c) A fault-propagation fold forms hinterland-ward of the triangle structure by a thrust ramp (2) modelling done with tri-shear. The model mimics the structure imaged in the seismic approximately from the deformation front (east) to the red stippled line.**





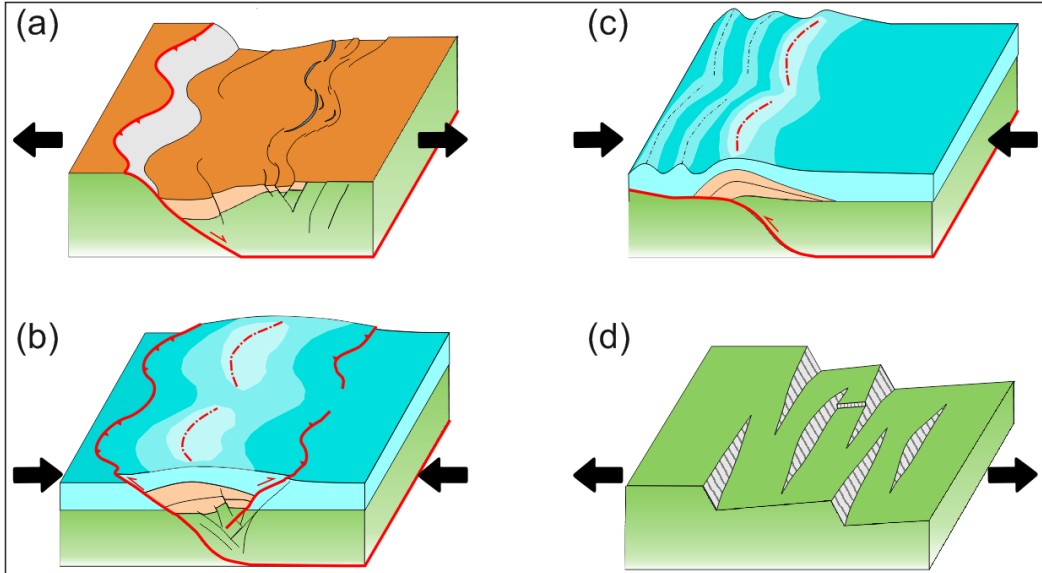

**Figure 11: (a) and (b) Model of extension with subsequent inversion on curved linked faults (modified after Yamada and McClay, 2004). (c) Adding a thin-skinned element to the sketch of inverted curved linked fault system (d) Sketch of half graben systems with overlapping faults for anticipation of more complex subsurface geometries before inversion.**



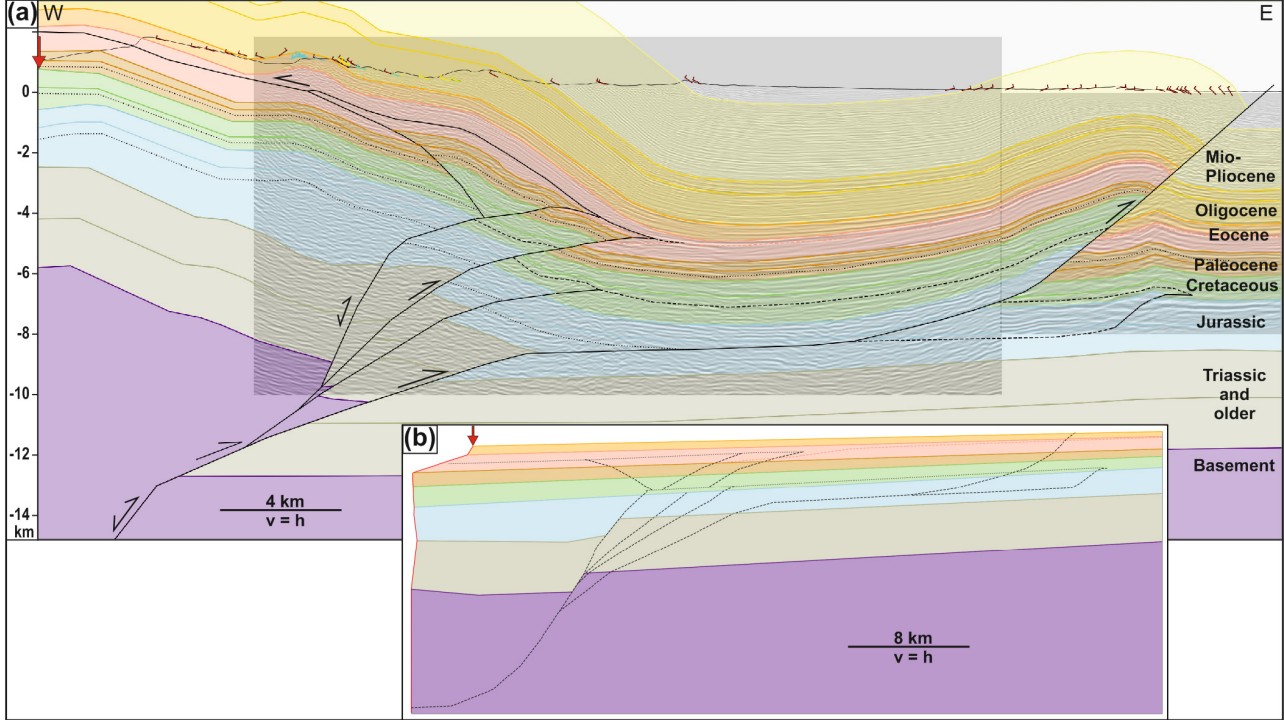

**Figure 12: (a) Constructed W-E section in the southern sector of the study area with PSDM seismic in the background. The section is balanced between the red lose line and the eastern end of the section (fixed line), KE: Kirthar Escarpment; (b) restored section (50% scale of (a)). Calculated shortening is approx. 10 km or 20%.**



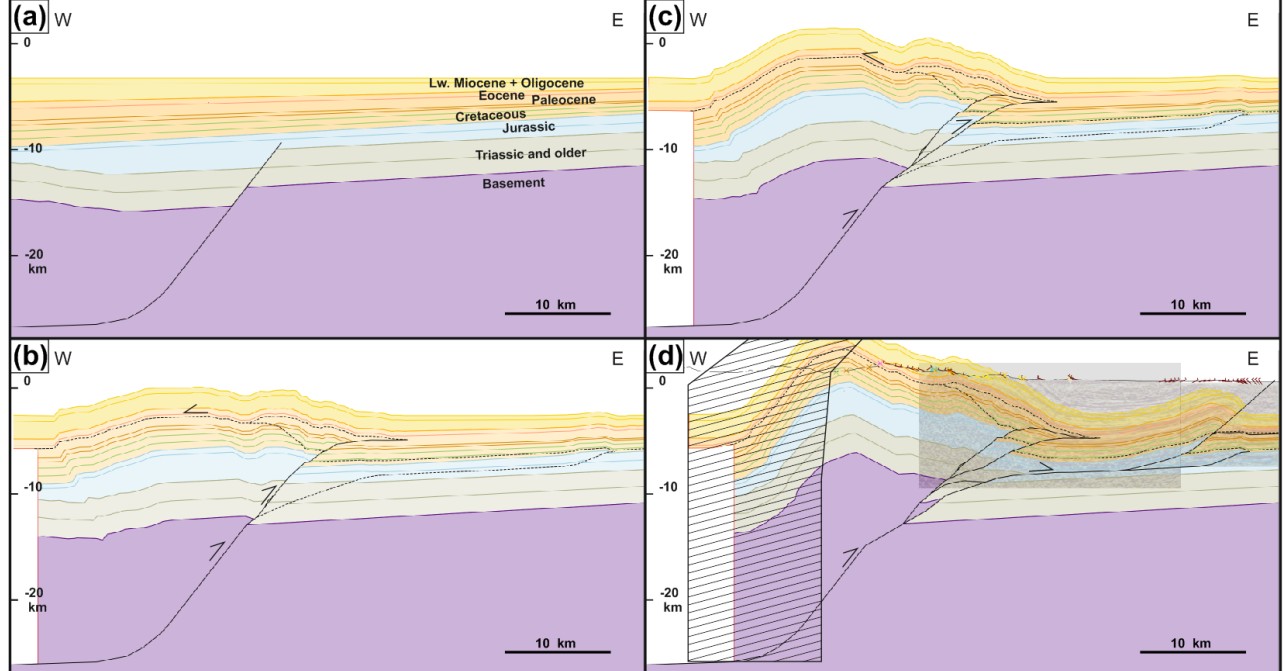

**Figure 13: Simplified kinematical evolution of the southern sector. (a) Pre-contractional situation with Jurassic normal fault. Thin stippled lines indicate faults of dominant layer parallel shortening, (c) and (b) Incremental deformation of imbrication and passive roof thrusting above the inverting normal fault. (d) Final geometry of the kinematical forward model compared to seismic and surface geology. The geometry in the hatched area in the western part of the section does not fit the surface geology and would require additional deformation by inverting faults and cover sediments not regarded in this model.**



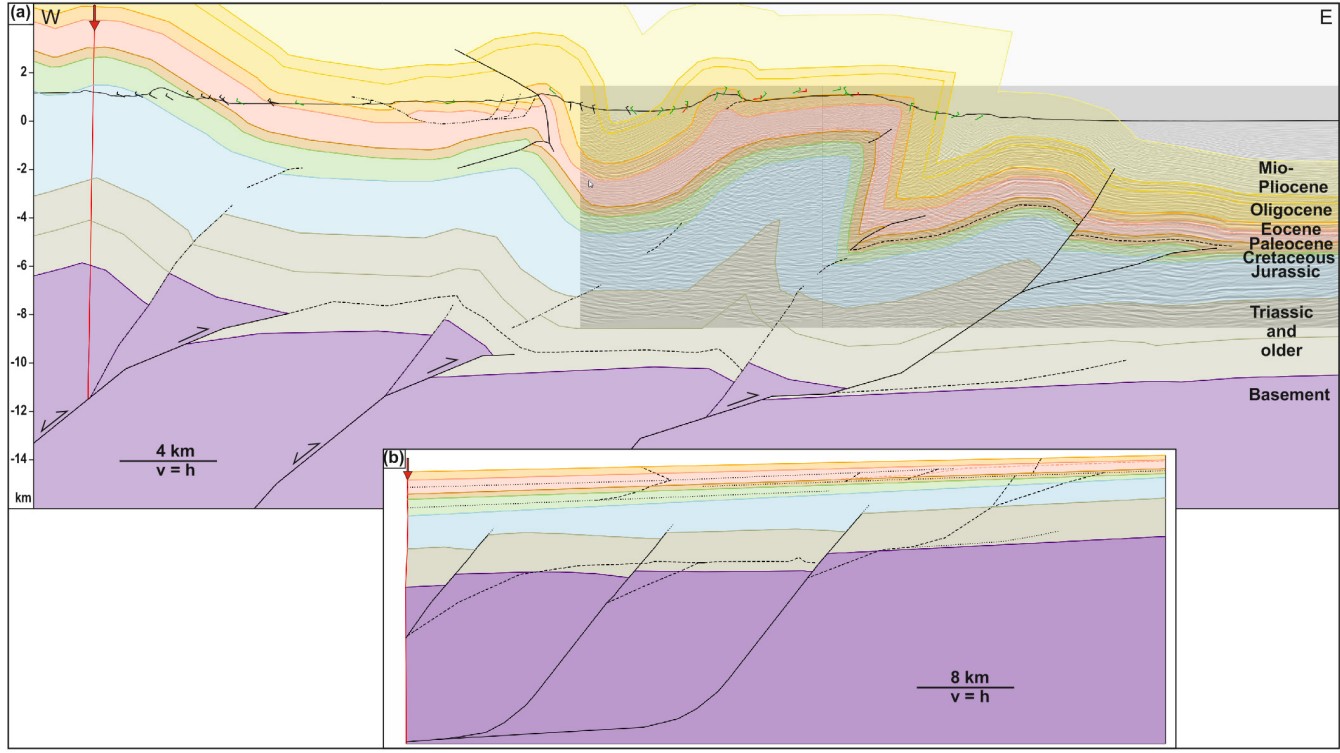

**Figure 14: (a) Constructed W-E section in the northern sector of the study area with PSDM seismic in the background. The section is balanced between the red lose line and the eastern end of the section (fixed line). (b) Restored section (at 50% scale of (a)) by using line length and area balancing methods. Calculated shortening is approx. 11.2 km or 18%**

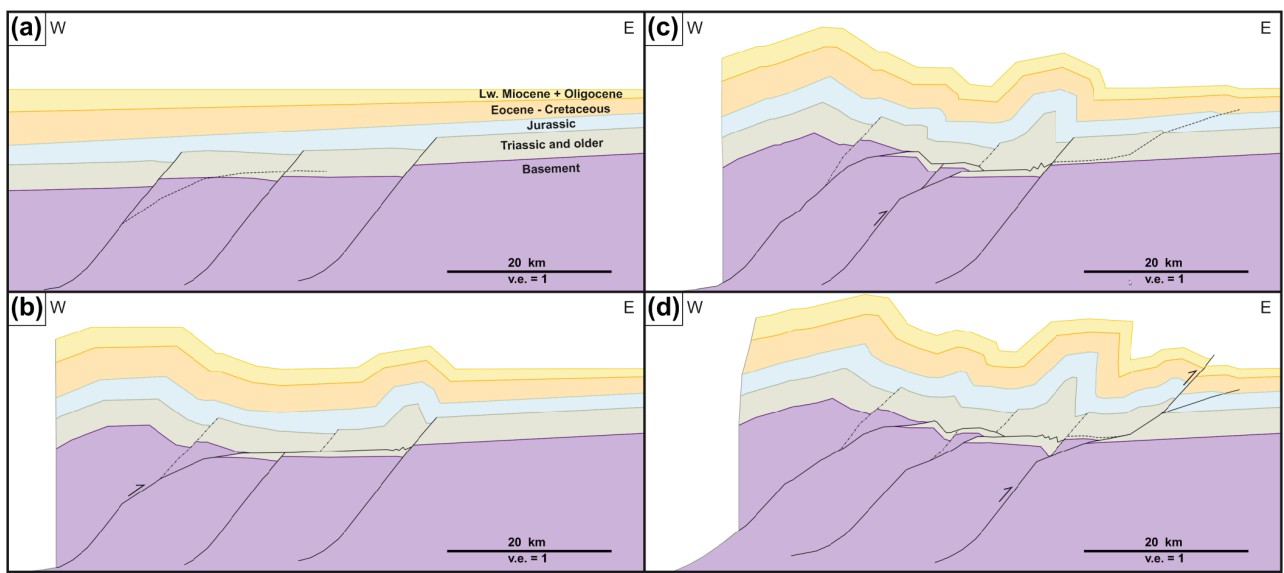

**Figure 15: Simplified kinematical evolution of the southern sector, shown for Eocene to Basement rocks. (a) pre-contractional situation with Jurassic normal faults. Stippled line indicated future shortcut fault. (b) and (c) Increments of inversion with shortcut faulting and detachment folding related to buttressing. (d) Final geometry of the kinematical forward model.**





| Figure Label | Event Id | Mw | Z | Date | Author | Dip-azimuth | Dip | Strike | Rake | Dip-azimuth | Dip | Strike | Rake |
|---|---|---|---|---|---|---|---|---|---|---|---|---|---|
| F1a | 603867342 | 5.4 | -10100 | 19.12.2013 | NEIC | 253.8 | 14.7 | 163.8 | 70.9 | 93.5 | 76.2 | 3.5 | 94.9 |
| F1b | 603867342 | 5.4 | -12000 | 19.12.2013 | GCMT | 291.0 | 39.0 | 201.0 | 133.0 | 61.0 | 63.0 | 331.0 | 61.0 |
| F1c | 603867342 | 5.4 | -12000 | 19.12.2013 | NEIC | 277.0 | 26.0 | 187.0 | 99.0 | 87.0 | 64.0 | 357.0 | 85.0 |
| F2 | 308027 | 5.4 | -15000 | 21.01.1992 | HRVD | 306.0 | 48.0 | 216.0 | 104.0 | 105.0 | 44.0 | 15.0 | 75.0 |
| F3 | 301671 | 4.9 | -15000 | 28.03.1992 | HRVD | 272.0 | 57.0 | 182.0 | 89.0 | 93.0 | 33.0 | 3.0 | 91.0 |
| F4 | 259589 | 5.1 | -33000 | 28.12.1992 | HRVD | 263.0 | 33.0 | 173.0 | 32.0 | 146.0 | 73.0 | 56.0 | 119.0 |
| F5 | 13436558 | 5.0 | -12000 | 17.03.2009 | GCMT | 304.0 | 45.0 | 214.0 | 106.0 | 102.0 | 47.0 | 12.0 | 75.0 |
| F6 | 604543379 | 5.0 | -12000 | 08.05.2014 | GCMT | 212.0 | 42.0 | 122.0 | 69.0 | 59.0 | 52.0 | 329.0 | 108.0 |

**Table 1: Nodal planes from International Seismological Centre (2015), reviewed events only. In addition to the dip the dip-azimuth of the planes is calculated. The strike and rake values from the database are given for completeness and assessment of obliquity. Event F1 has 3 different solutions in the database. The differences are a rough indication of the uncertainty of the data. Green/red dip values indicate the lower and higher dip surfaces of the pair.**