# Peer review of "Linked thick to thin – skinned inversion in the central Kirthar Fold Belt of Pakistan"

_Solid Earth, 2018_

## Referee Comment (RC1) · Anonymous Referee #1 · 7 Jan 2019

This paper uses a combination of seismic reflection profiles, field observations, and structural modelling to examine the central part of the Kirthar fold belt in western Pakistan. The main result is that the region represents a combination of thick-skinned reactivation of normal faults and thin-skinned deformation of the overlying sediments. At present, I think significant clarifications will be necessary in order to make the paper suitable for publication, as described below.

1. As a general point, I struggle to see, and judge the robustness, of some of the interpretations of the seismic data (e.g. the images in figure 8). I think that it would be helpful if the authors: (a) provided un-interpreted, as well as interpreted, images of the sections; (b) provided zooms of the key features discussed in the text; (c) marked on the locations of the wells that seem to be key to the correct identification of some

horizons, (d) provide in the text a discussion of the reasoning behind interpreting the structures, and the horizon identifications. For example, at the eastern ends of the seismic sections, and in places where it is buried, where the pick of the top of the Kirthar formation is key to the subsequent discussion of regional level, a detailed discussion about the reasoning of the pick would be useful. On some seismic sections I can't tell what the basis is for interpretations (e.g. faults 1 and 2 on fig 9). In general, I think a much more thorough analysis and justification of the seismic data is necessary.

2. I think the authors would benefit from a clearer consideration of the seismicity. Focal mechanisms are provided in Figure 2, but they are wrongly attributed (the ISC only estimates locations, not mechanisms, so these mechanisms must be sourced from elsewhere). The depth of these events is not discussed (i.e. are they within the deformed sedimentary sequence, or the underlying basement?). I think the authors would benefit from searching the literature for well-constrained locations, mechanisms, and depths for earthquakes in this region, and discussing the relationship between the geometry of the active faulting and the structural models they propose. In addition, Ambraseys and Bilham (Bulletin of the Seismological Society of America, Vol. 93, p. 1573–1605, 2003) contains much useful information on the historical seismicity, including the 1931 event close to the Krithar range-front, which will have important implications for the kinematics of the shortening. Some of the arguments based upon the dip of the faulting (e.g. that thrust faults can't form at dips of 45 degrees; section 5.1) are incorrect, based on observations of faults this steep being newly-formed in oceanic outer rises (e.g. Craig et al, EPSL, 392, 94-99, 2014).

3. Little detail is given of the structural reconstructions (Figs 12-15). For example, what is the justification behind each step in the reconstruction, and how many other interpretations are possible which match the observations? The authors acknowledge that the solution is not unique, but I have little feel for how many different configurations are possible, why these models were chosen, and how alternative models would affect their conclusions. I think these issues need to be discussed in detail (particularly the

final one), and I think that for each stage in the reconstructions a reason should be given for why that deformation has been chosen (e.g. in order to match feature X, we now need to undertake deformation Y).

4. Although I can see why the authors have suggested a combination of thick- and thin-skinned deformation in this region, it's not clear to me why this definitely needs to be the case, rather than just one of a range of possibilities. The pattern of folding is described as being analogous to an array of normal faults, but I don't see why this necessarily needs to be the case – the folding looks fairly similar to that in the Zagros mountains, where it is thought that the folds are decoupled from the underlying basement by the Hormuz salt. The second paragraph of section 5 simply states their view, without justifying it. For example, how have they ruled out the possibility of more thickening in the deeper parts of the sedimentary layer in the western parts of the section giving the change in structural level? Given the thickness of the sediments, this seems equally plausible? If the authors are going to pick a preferred viewpoint, I think they need to give a detailed justification.

5. In general, I think the manuscript would benefit from many of the statements being backed up with observations and/or reasoning. For example, in section 4 the lateral thickness change in the Ghazij Shales is stated. However, we are not told what information this was based on (i.e. where are the well or surface observations, or how is the top and bottom of this unit recognised in the seismic data). This thickness change is key to their suggestion of the reactivation of normal faults. There are many statements like this in the text, which leave the reader wondering what the conclusion is based upon. I think it would be very helpful to the reader if the authors provided supporting logic or observations of all statements they make.

---

## Referee Comment (RC2) · Kley (Referee) · 15 Jan 2019

General comments:

R. Hinsch and colleagues present a rather straightforward structural analysis and interpretation of the central Kirthar Range thrust front in Pakistan. Their study is based on (partly new) seismic data, DEMs, satellite imagery, and some field work. The main point they are making is that a narrow belt of thin-skinned deformation at the thrust front is linked to basement-involved thrusts in the internal parts of the thrust belt. These thrusts or reverse faults are interpreted to result from the reactivation of normal faults inherited from the time when India rifted from other parts of Gondwana. I find no fundamental issues with the ideas and the way they are presented.

[Figure]

Specific comments:

In the way of data, the weakest part of the paper is definitely the claim that reactivated normal faults are involved in the deep structure. The seismic profiles do not reach deep enough to show anything conclusive. The earthquake nodal planes except one at 57° dip too gently to satisfy the Coulomb-Mohr prediction for normal faults. In fact, the average (arithmetic mean) dip angle of the west-dipping nodal planes is only 38°, much closer to an ideal Mohr-Coulomb thrust fault than normal fault. The normal faults of the structural model dip around 50°. Judging from the stratigraphic description and the authors' comments, the timing of active rifting isn't very well constrained, either. The same seems to hold true for the depth to and nature of the basement. I therefore recommend to tone down the inversion-related part of the interpretation while maintaining that the basement must be involved in thrusting.

When the authors compare their new structural models to Banks and Warburton's passive-roof duplex interpretation they should at least briefly discuss what happens in the more internal parts of the belt, away from the deformation front. The passive-roof model was motivated by the need to explain gently folded strata raised well above the regional level for a considerable across-strike distance. I assume that this problem also applies to the central Kirthar Range. If the Kirthar Range is held up by a series of reactivated normal faults, where is the reverse displacement of the more internal faults accommodated that cannot be transferred to the thin-skinned front? Or, in other words, is there enough shortening in the internal Kirthar Range to support its topographic and structural elevation assuming that the basal detachment is in the basement?

I am not entirely convinced by the uniqueness of the sequence of thrusting derived in Fig. 9. The advance of a thrust wedge between thrust 1 and backthrust 2 would result in kink band migration and not "progressive limb rotation" as described by the authors (l. 22 in text) and actually suggested by the growth strata geometries. It is also interesting that the kink axis shown to be associated with the tip of the wedge at deeper levels appears offset to the west in the growth strata, but also in the syncline suggested below thrust 1. I could imagine a scenario with no bedding-parallel backthrust and thrust 1 as a late subhorizontal structure displacing the syncline axis towards the east. The implication would be that there must be another thin-skinned thrust further east.

One thing I am deeply skeptical about is the landslide interpreted in Fig. 6 b. The way this feature is described in the caption I gather that it is supposed to have formed by draping over the topography of the steep forelimb (or did I get that wrong?). I find it hard to believe that you could form the orderly anticline depicted in the satellite image from a rock mass sliding over an irregular land surface. I think that the paper would strongly benefit from a few additional figures. First, it would help the imagination to have a regional cross-section reaching west to the strike-slip system. Secondly, I strongly recommend to prepare a synthetic figure that combines the new cross-sections with those from published studies whose locations are shown in Fig. 1, preferrably redrawn such that comparison is made easy. Nobody wants to look up four other papers to see what the paper they are presently reading is talking about.

Technical: See annotated pdf file in supplement

Please also note the supplement to this comment:
https://www.solid-earth-discuss.net/se-2018-137/se-2018-137-RC2-supplement.pdf

―――――――――――――――――

**Supplement:**

[revised manuscript text omitted]

---

## Referee Comment (RC3) · Kley (Referee) · 15 Jan 2019

Kley (Referee)

jonas.kley@geo.uni-goettingen.de

I just realized that the first supplement I uploaded contains quite a number of un-commented text marker highlights. Some of those were just marks I left for my own attention during a first perusal. They are deleted from the new supplement. For others, I have added the comments. Remaining uncommented marks refer to missing blanks, missing hyphens, erroneous capitalization and so forth. Some indicate issues I already marked and commented on further up in the MS.

Please also note the supplement to this comment:
https://www.solid-earth-discuss.net/se-2018-137/se-2018-137-RC3-supplement.pdf

[Figure]

**Supplement:**

[revised manuscript text omitted]

---

## Author Comment (AC1) · 20 Feb 2019

**Reply to RC1 by Anonymous Referee #1**

We thank the anonymous Referee for reviewing the manuscript. Most of the comments ask for clarifications of interpretations and assumptions. These comments give us the opportunity to substantially improve the readability of the manuscript. Some comments we do not fully agree or we think they are not relevant. All points raised will be discussed below.

**1a) RC1:**
"As a general point, I struggle to see, and judge the robustness, of some of the interpretations of the seismic data (e.g. the images in figure 8). I think that it would be helpful if the authors: (a) provided un-interpreted, as well as interpreted, images of the sections;"

**Response:**
a) Figure 8 was composed in a way that as little of the seismic is covered by interpretation and labelling so the reader can get an impression on the (often poor) quality of the seismic image. We followed the example given by other papers in Solid Earth using nearly un-interpreted seismic with some labelling to highlight features discussed in the text (e.g. Malehmir et al. 2018, Tavani et al., 2018, Gallastegui et al. 2016). The interpreted version of the seismic is given in Figures 12a and 14a where one can see the complete interpretation (the semi-transparent seismic in the background). We added cross-references between the figures to clarify this, thus enabling the reader to compare the final structural interpretation and original seismic. Because of the different length of the sections (Seismic in Fig 8 and the final, extended sections in Figs 12a, 14a), it seems unreasonable to combine the figures. Instead, in order to provide an unbiased documentation on the database, we added a supplementary figure that shows the seismic of Figure 8 in high resolution without any annotations at all.

**Changes to the MS**
Supplementary high resolution figure with blank seismic (Supplementary Figure 2).

Improved cross-references between figure captions (Figs. 12a and 14a to Fig. 8) and text (see track changes MS).

**1b) RC1:**
"b) provided zooms of the key features discussed in the text;"

**Response:**
We do not agree that one would be able to see more details in zooms of the seismic. The size of Fig. 8 is limited by the column width of the journal but it already resolves all the features visible in the seismic which are referred to in the text (as they are indicated by small letters). The PDF version of the MS for review might be lacking the quality of the figure expected. We think by providing a large scale high resolution version of the blank seismic (Supplementary Figure 2) we sufficiently provide documentation of the data our interpretation is based on. The seismic won't get better in zoom figures. We will make also sure, that the final submitted version of Fig. 8 will have as high quality as reasonably possible.

**Changes to the MS**
Providing a high resolution blank seismic of Figure 8 in the supplementary material (Supplementary Figure 2).

A high resolution blank seismic of Figure 9 (Supplementary Figure 3) will be provided as well.

"c) marked on the locations of the wells that seem to be key to the correct identification of some horizons"

**Response:**

This comment likely asks for the exact location of the wells on the seismic?
For confidentiality reasons we cannot give exact well locations. But we edited the text and figures to improve the MS accordingly.

**Changes to the MS**

We improved the description of the stratigraphic control and well locations in the text (Section 4). We added seismic horizon labelling to Figure 8 with indications in the stratigraphic column (Fig. 3). Better explanation of well control location in Figure 8. Please see track changes MS.

**1d) RC1:**

"(d) provide in the text a discussion of the reasoning behind interpreting the structures, and the horizon identifications. For example, at the eastern ends of the seismic sections, and in places where it is buried, where the pick of the top of the Kirthar formation is key to the subsequent discussion of regional level, a detailed discussion about the reasoning of the pick would be useful. On some seismic sections I can't tell what the basis is for interpretations (e.g. faults 1 and 2 on fig 9). In general, I think a much more thorough analysis and justification of the seismic data is necessary.

**Response:**

The horizon interpretation is done based on the well control and seismic facies. As mentioned above seismic horizons as depicted from the well control have been added to Fig. 8 and are explained in Fig. 3. Additionally the seismic characteristics of the Kirthar limestone pick is now given in Section 4.
For the fault interpretation in Figure 9 we added also the non-interpreted seismic as supplementary figure. In addition, a new figure with a kinematic scheme is provided and explained, highlighting the rationale behind the interpretation (this is also part of the answer to RC2).
We think that an interpretation like Fault 1 does not require a detailed justification. Tilting and uplifting strata from a horizontal position the way it is imaged requires a fault. Such basic interpretations follow well known structural concepts, cf. AAPG Atlas of Shaw et al., 2005). The fault 2 (roof thrust) is a geometrical necessity if no further deformation occurs to the east. This rationale is now better explained by the new figure and the description of the missing deformation towards the east in the revised MS.

**Changes to the MS**

Additions to section 4: Stratigraphic control by wells, seismic grids and seismic characteristics of Kirthar pick.
Additions of the picked horizons to stratigraphic column (Fig. 3) and the eastern part of the seismic lines (Fig 8).
Improved Figure 9 with a kinematic scheme and rewording of the regarding text in Section 4.1.
Supplement Fig. 3: Seismic of Fig. 9 without interpretation

**2) RC1:**

"I think the authors would benefit from a clearer consideration of the seismicity. Focal mechanisms are provided in Figure 2, but they are wrongly attributed (the ISC only estimates locations, not mechanisms, so these mechanisms must be sourced from elsewhere)."

**Response**

Focal mechanisms were downloaded from ISC database
([http://www.isc.ac.uk/iscbulletin/search/fmechanisms/](http://www.isc.ac.uk/iscbulletin/search/fmechanisms/)). We followed the citation scheme
proposed on the ISC webpage. The contributing agencies to the ISC database that we used
were actually listed in Table 1 in the author column. See also Lentas et al. (2018, "The ISC
Bulletin as a comprehensive source of earthquake source mechanisms")

**Changes to the MS**

We clarified the contribution of other agencies to the ISC database and added the reference
"Lentas et al. (2018)" to the caption of Table 1

**2 contd.) RC1:**

"The depth of these events is not discussed (i.e. are they within the deformed sedimentary
sequence, or the underlying basement?)".

**Response:**

This is a good point that is missing in the manuscript. We were aware of the depth and
considered the depth of the events when interpreting the data (as "Z" is listed in Table 1).

**Changes to the MS**

We added a consideration of the depth of the events to Section 5.2 (former 5.1). See track
changes document.

**2 contd.) RC1:**

"I think the authors would benefit from searching the literature for well-constrained locations,
mechanisms, and depths for earthquakes in this region, and discussing the relationship
between the geometry of the active faulting and the structural models they propose. In
addition, Ambraseys and Bilham (Bulletin of the Seismological Society of America, Vol. 93,p.
1573–1605, 2003) contains much useful information on the historical seismicity, including the
1931 event close to the Krithar range-front, which will have important implications for the
kinematics of the shortening"

**Response**

We struggle to find significant other sources of focal mechanisms relevant to our study area.
We consider the ISC Bulletin database with approx. 150 contributing agencies to be the most
relevant data source for this study. A dedicated seismological study for the study area would
certainly help, but is beyond the scope of this study.

We are of the opinion, that the distribution of seismic events (i.e. the depth distribution) does
not add value to the discussion about fault geometries and kinematics. The figure below
shows recorded seismic events projected (+- 50 km perpendicular to the section plane) on
the extended section of the new regional section (Fig. 16g). Data from the ISC bulletin partly
line up in certain depth which could be related to poor depth location/artefacts. Improved
locations of the EHB database (using algorithms after Engdahl et al. 1998) do not improve
the picture, but actually show that the data in general has a large scatter.

[Figure]

The paper by Ambraseys and Bilham (2003) does not provide focal mechanisms but, as suggested by the reviewer, a discussion on fault kinematics on the 1931 Mach Event (based mainly on levelling data). Using the same levelling data the potential fault shape and kinematic is evaluated also in Szeliga et al. (2009). Those authors show focal mechanisms on their Fig. 1 after Harvard MCT, a source that is included in the ISC Bulletin database that we used for our Fig. 2.

We follow the suggestion by the referee and introduce the work done on the Mach 1931 Earthquake in Section 5.2. The geological section of Szeliga et al. (2009) with the approx. fault shape of the Mach 1931 event has been added as Figure (Fig. 16d). The fault shape considered by Szeliga et al. 2009 is similar to the frontal fault system in our study area. The consequences are also now discussed in Section 6.2.

**Changes to the MS**
We added the citations and a description of the results of Ambraseys and Bilham (2003) and Szeliga et al. (2009) to Section 5.2 (former 5.1). The geological section given in Szeliga et al. 2009 has been added to a new Figure 16 (Figure numbers will be resorted in final revision), which shows a compilation of sections (in response to RC2). The fault responsible for the 1931 Mach Event as suggested by Szeliga et al. 2009 is also shown in Figure 16d. The consequences /relationship of the deformation is additionally discussed in Section 6.2. See track changes document.

**2 contd.) RC1:**
"Some of the arguments based upon the dip of the faulting (e.g. that thrust faults can't form at dips of 45 degrees; section 5.1) are incorrect, based on observations of faults this steep being newly-formed in oceanic outer rises (e.g. Craig et al, EPSL, 392, 94-99, 2014),".

**Response**
It has not been stated in the reviewed MS that thrusts cannot form at dips of 45°. We say, we consider the steeper faults (i.e. 45° and more) too steep to represent newly initiated faults. That is not exactly the same. The reason, why we think they are not newly initiated faults is based on a line of arguments, which includes fault and fold orientation in respect to the plate kinematic direction. Section 5 and sub-sections 5.1-5.3 have been extended to clarify the reason for our interpretation of inversion (also in respect to RC2)
We consider that the paper by Craig et al. (2014), suggested by the Referee 1, is not relevant in the discussion whether thrust faults can form at higher angles or not. In the paper Craig et al. argue that they observe normal faults forming not at an ideal angle of 60° but cluster around 45° (in oceanic crust). In order for that to happen, they imply that the rocks must have a lower than usual coefficient of friction. They speculate that the suspected low coefficient of friction is a result of hydrothermal alteration of the oceanic crust after it formed at the MOR. Craig et al. also plot histograms of nodal planes from thrust faults which show clusters above and below 45°. However, the histogram shows both nodal planes, so the cluster on the higher angles could represent the auxiliary plane – or reactivated faults. Craig et al. do not suggest that thrust faults form above a certain angle and they explicitly do not analyse the thrust faults (Craig et al. (2014):*"The population of thrust-faulting earthquakes (Fig.3C) is too small for any clear trends to emerge, and is not the subject of further analysis in this study"*). Consequently, their results are only valid for normal faults. Furthermore, the line of arguments that is used for normal faults forming a lower angles seem not to fit for thrust faults forming at steeper angles than usual. If the coefficient of friction is lower than for standard Andersonian faults (i.e. lowered by a hydrothermal processes), thrust faults would form at lower angles, not higher. For thrust/reverse faults to form at higher angles the coefficient of friction would need to be higher than the normally considered used value (i.e. 0.6 for 30° thrust fault, all needed references are given in Craig et al., 2014). Consequently,

we interpret reverse faults with dips at 45° or above indicate rather frictionally reactivated faults than newly initiated faults.

**Changes to the MS:**
Section 5.2 (former section 5.1): "We interpret these steep faults therefore as parts of pre-existing faults that are in a suitable angle for reactivation".
A suggestion on partial fault reactivation and other inversion related deformation in Section 5.3. (marked as reply to RC2). See track changes document.

**3) RC1:**
"Little detail is given of the structural reconstructions (Figs 12-15). For example, what is the justification behind each step in the reconstruction, and how many other interpretations are possible which match the observations? The authors acknowledge that the solution is not unique, but I have little feel for how many different configurations are possible, why these models were chosen, and how alternative models would affect their conclusions. I think these issues need to be discussed in detail (particularly the final one), and I think that for each stage in the reconstructions a reason should be given for why that deformation has been chosen (e.g. in order to match feature X, we now need to undertake deformation Y)".

**Response**
Strictly speaking Figures 13 and 15 do not show reconstructions but simplified kinematical forward models. Reconstructions are the restored sections (Fig. 12b and 14b). The techniques how Figs. 12b and 14b are restored are defined in the text (Section 5.4, last paragraph) and follow established procedures (e.g. Woodward et al., 1989). The simplified kinematical forward models in Figs. 13 and 15 are suggestions that show that the restored section also is meaningful in a kinematical sense. The necessity to link restored and present day stages are the main constraints. We clarified this relation in the revision. Some additional reasoning for the individual steps have been added as well.
The question on "how many different configurations are possible, why these models were chosen, and how alternative models would affect their conclusions" is not simple to answer. We follow in the MS the approach of using as many constraints as possible (surface geology, seismic, well data, regional setting, the nodal plane geometries, balancing constraints and regional elevation considerations etc.) and combining them in a logical way in order to shrink the amount of admissible solutions. How much change on our solution is a new solution or just an adaption in the frame of the given solution is a matter of definition and also scale dependent. We consider that the involvement of basement deformation can be considered as certain. That these are likely inverting normal faults (or part of it) is considered as very likely, based on a thread of arguments (which is now elaborated more clearly in Section 5.1. -also in respect to the comments by Referee 2). If these faults are of original Triassic or Jurassic age remains relatively uncertain. The same applies for the amount and exact shape of faults in the subsurface. Our main conclusion is based on the solution which we consider almost certain. It would require some very good ideas to combine all the constraints and come up with a solution that is different from being just a modification of our model. However, learning about potential other solutions has not only a scientific but also a business impact, so we encourage substantial alternative explanations that contradicts our main conclusions. By documenting the database as good as confidentiality allows, we hope to serve this purpose.

**Changes to the MS:**
The mentioning of the balanced sections and restorations (Figs. 12 and 14) moved up in the text (now in Section 5. Second paragraph). By this we can refer to the restored and balanced section as start and finite stage respectively when explaining the kinematic models (Fig. 13 and 15), improving the context/readability. Additionally some improved reasoning for the chosen steps, as suggested by the Referee1, have been added (see track changes MS).

To section 6.5 we added: "How much change on our model is a new solution or just a modification is a matter of definition and also scale dependent."

**4) RC1:**
"Although I can see why the authors have suggested a combination of thick- and thin- skinned deformation in this region, it's not clear to me why this definitely needs to be the case, rather than just one of a range of possibilities. The pattern of folding is described as being analogous to an array of normal faults, but I don't see why this necessarily needs to be the case – the folding looks fairly similar to that in the Zagros Mountains, where it is thought that the folds are decoupled from the underlying basement by the Hormuz salt. The second paragraph of section 5 simply states their view, without justifying it. For example, how have they ruled out the possibility of more thickening in the deeper parts of the sedimentary layer in the western parts of the section giving the change in structural level? Given the thickness of the sediments, this seems equally plausible? If the authors are going to pick a preferred viewpoint, I think they need to give a detailed justification."

**Response:**
"The second paragraph of section 5 simply states their view, without justifying it."
Manuscript: "We suggest that the order of structural uplift (larger than 5500 m) is linked to a deeper structural level in the basement." The order of uplift is used as an temporary justification. The sentence is part of a paragraph that is a header for the complete Section 5 (including subsections) in which the reasoning for thick-skinned contribution is elaborated and more reasoning/justification is given. We improved the wording to make this clear. Subsection 5.1 in combination with a new Figure 17 now addresses the question why a thin-skinned solution (thickening in deeper parts) is unlikely. This builds up on a newly added regional section (Fig. 16g) plus an overview geological map (new background in Fig. 2) which has been added as part of the response to RC2. The main reason is that a duplexes would cause severe balancing issues and are also not likely in the transpressional setting that does not seem to work like a classical accretionary wedge (details in the Track changes MS).
"the folding looks fairly similar to that in the Zagros mountains, where it is thought that the folds are decoupled from the underlying basement by the Hormuz salt"-
When comparing these regions we probably need to limit the similarity of the complex fold pattern and double plunging folds to the southern/southeastern Fars Arch of the Zagros, where Hormuz salt is present as detachment (cf. Bahroudi and Koyi, 2003). The deformation style in the Simply Folded Belt along strike the Zagros is not everywhere the same (cf. Allen and Talebian 2011). Nevertheless, the main differences are: in the southern Fars area most of the folds return to regional elevation (or close to it) in the trailing synclines (as evident on sections and geological maps, e.g. Jahani et al., 2009). The folds are large scale detachment folds influenced by the halokinetic evolution (requiring diapirs) since the Paleozoic (e.g. Jahani et al. 2009, Callot et al. 2012). The double plunging fold shapes in the SE Fars are a result of complex interaction of halokinetic induced stratigraphic thickness variations and shortening on a salt detachment with a likely not planar detachment plane. The role of basement involvement in the deformation in the Fars has been proposed (e.g. Jackson 1980) and is debated since. For the SE Fars region a strong gain of structural elevation is evident only towards the hinterland in the imbricate zone (behind the High Zagros Fault cf. Fig. 2 of Mouthereau et al, 2007, or Fig. 2 of Bahroudi and Koyi, 2003). This faults thus likely marks a stepping down of the detachment into the basement.
In the Kirthar Fold belt there is no evidence of salt presence in the stratigraphy and no evidence for salt tectonics. The second most important difference is that all rocks west of the Kirthar Escarpment are significantly elevated above their regional elevation of the undeformed foreland (more than 6000 m).

Although the comparison to the Fars Arch might be interesting, we do not think that it would improve the MS.

**Changes to the MS:**
Added Figure 16g (regional section), Figure 16h (average topographic profile along 16g) and new background in Fig. 2 (overview geological map) also as reply to RC2.
Added Figure 17 schematic scheme for discussion of structural elevation uplift
Section 5.1 now addresses the question why a thin-skinned solution is less likely to explain the structural elevation gain towards the west.
Section 5.1 also includes a discussion why we consider a thick-skinned (not inversion related) deformation less likely also (in response to RC2). See track changes document.

**5) RC1:**
"In general, I think the manuscript would benefit from many of the statements being backed up with observations and/or reasoning. For example, in section 4 the lateral thickness change in the Ghazij Shales is stated. However, we are not told what information this was based on (i.e. where are the well or surface observations, or how is the top and bottom of this unit recognised in the seismic data). This thickness change is key to their suggestion of the reactivation of normal faults. There are many statements like this in the text, which leave the reader wondering what the conclusion is based upon. I think it would be very helpful to the reader if the authors provided supporting logic or observations of all statements they make."

**Response:**
We do not agree that we are using un-backed up statements in the MS. Unfortunately, there is only one example given in RC1, which we think is not fully adequate: RC1: "For example, in section 4 the lateral thickness change in the Ghazij Shales is stated. However, we are not told what information this was based on (i.e. where are the well or surface observations, or how is the top and bottom of this unit recognised in the seismic data)". The manuscript reads in Section 4.1, page 8 Line 3-5: *"The low-reflectivity seismic facies below the Kirthar limestones are Eocene Ghazij Shales (Fig. 8a, point g). These shales thicken dramatically from wells in the East (several tens of meters) towards the West (several hundreds of meters, constrained by outcrop and seismic velocity data)"*. We actually would consider this as an explanation and not an un-backed up statement. The location of the gas condensate fields on the frontal anticline has been described in Section 4. The relative reference where the location is on the section was given). The outcrop situation of the Ghazij can be checked on Figure 5 and has been described in Section 3.1. However, we admit, that this might not be easy for the reader to follow. So, for convenience we improved the description of this thickness increase and the observations where it is based on. Well control and seismic interpretation description has been improved (as described in points 1c and 1d of this reply)
RC1:" This thickness change is key to their suggestion of the reactivation of normal faults."
This is actually not the case. The thickness variations in the Ghazij Formation is nowhere used in the MS as argument for normal fault reactivation.
A thick Ghazij Formation, however, is considered as suitable roof thrust. The suitability as weak layer has been already demonstrated in Section 3.1/Fig. 6.
The changes in respect to the other comments from RC1 (above) and also in respect to RC2 should have significantly improved the MS and allow the reader to follow our interpretations and conclusions (without "*non-back-upped statements*").

**Changes to the MS:**
Section 2 Thickness trend description of Ghazij shales in Tectonostraigraphic evolution
Section 3.1.: These shales reach several hundred meters of thickness east of the Kirthar Escarpment.

Section 4: improved description on well control and stratigraphic interpretation (including annotations of stratigraphy in Fig. 8)

Section 4.1 Changed the description of the thickening shales to: "These shales thicken dramatically from the wells on the frontal anticline in the East (several tens of meters) towards the West (several hundreds of meters, constrained by seismic velocities and outcrop information just west of the Kirthar Escarpment, cf. Fig. 5 and Ahmad et al. 2012)".

**References used in this reply not present in the reference list of the revised manuscript:**

Allen, M.B. and Talebian, M., Structural variation along the Zagros and the nature of the Dezful Embayment., Geol. Mag., 148 (5-6). pp. 911-924, 2011

Bahroudi, A., Koyi, H.A., Effect of spatial distribution of Hormuz salt on deformation style in the Zagros fold and thrust belt: an analogue modelling approach. J. Geol. Soc. Lond. 160 (5), 719-733, 2003

Callot, J.-P., Trocme, V., Letouzey, J., Albouy, E., Jahani, S., Sherkati, S., Preexisting salt structures and the folding of the Zagros Mountains. In: Alsop, G.I.,Archer, S.G., Hartley, A.J., Grant, N.T., Hodgkinson, R. (Eds.), Salt Tectonics, Sediments and Prospectivity, vol. 363. Geol. Soc. London Spec. Pub., pp. 545-561. 2012

Craig, T. J., Copley, A and Middleton, T.A., Constraining fault friction in oceanic lithosphere using the dip angles of newly-formed faults at outer rises. Earth Planet Sc Lett, 392:94–99, 2014b. doi: 10. 698 1016/j.epsl2014.02.024, 2014

Engdahl, E.R., R. van der Hilst, and R. Buland. Global teleseismic earthquake relocation with improved travel times and procedures for depth determination, Bull. Seism. Soc. Am. 88, 722-743. 1998

Gallastegui, J., Pulgar, J. A., and Gallart, J.: Alpine tectonic wedging and crustal delamination in the Cantabrian Mountains (NW Spain), Solid Earth, 7, 1043-1057, https://doi.org/10.5194/se-7-1043-2016, 2016

Jahani, S., Callot, J.P., Letouzey, J., Frizon de Lamotte, D. The eastern termination of the Zagros Fold-and-Thrust Belt, Iran: structures, evolution, and relationships between salt plugs, folding, and faulting. Tectonics 28 (6), TC6004, 2009

Malehmir, A., Bergman, B., Andersson, B., Sturk, R., and Johansson, M.: Seismic imaging of dyke swarms within the Sorgenfrei–Tornquist Zone (Sweden) and implications for thermal energy storage, Solid Earth, 9, 1469-1485, https://doi.org/10.5194/se-9-1469-2018, 2018.

Mouthereau, F., J. Tensi, N. Bellahsen, O. Lacombe, T. De Boisgrollier, and S. Kargar, Tertiary sequence of deformation in a thin-skinned/ thick-skinned collision belt: The Zagros Folded Belt (Fars, Iran), Tectonics, 26, 2007

Shaw, J.H., C.D. Connors and J. Suppe Seismic interpretation of contractional fault-related folds. An American Association of Petroleum Geologists seismic atlas. American Association of Petroleum Geologists, Studies in Geology, v. 53, 2005

Tavani, S., Parente, M., Puzone, F., Corradetti, A., Gharabeigli, G., Valinejad, M., Morsalnejad, D., and Mazzoli, S.: The seismogenic fault system of the 2017 Mw 7.3 Iran–Iraq earthquake: constraints from surface and subsurface data, cross-section balancing, and restoration, Solid Earth, 9, 821-831, https://doi.org/10.5194/se-9-821-2018, 2018.

Woodward, Boyer and Suppe, Balanced Geological Cross-Sections: An Essential Technique in Geological Research and Exploration, Short Courses in Geology, Volume 6, American Geophysical Union, DOI:10.1029/SC006, 1989

---

## Author Comment (AC2) · 20 Feb 2019

**Manuscript file with coloured track changes**

**Changes in this colour are referring to changes regarding RC1.**

**Changes in this colour are done mainly in response to RC2**

[revised manuscript text omitted]

**4. Seismic interpretation and analysis**

The frontal-most anticline (Fig. 5) hosts several gas condensate fields and is partly covered by 2D seismic and at least one 3D seismic cube. From 2014-2017 two new 2D seismic surveys were acquired west of this frontal anticline. For confidentiality reasons we are unable to show exact locations of the seismic lines and the well data. However, we subdivided the area into a northern and a southern sector (Fig. 5) and use two representative W-E composed seismic sections to discuss the structural differences of these sectors (Fig. 8). The seismic surveys have up to 6 km horizontal spread and up to 240 fold and utilized dynamite as the source. Processing of the lines in Fig. 8 is  to Pre-Stack Depth Migration (PSDM). These seismic lines have been tied to the vintage seismic data and wells for stratigraphic control. Stratigraphic control on the lines is given by wells on the frontal anticline or in the foreland via a grid of vintage 2D lines or the 3D seismic cube. Based on this data robust grids of Oligocene to Cretaceous (Jurassic partly) horizons exist along the frontal anticline and the un-deformed foreland. Horizons are indicated in the un-deformed foreland in Figure 8 as well as in the stratigraphic column (Fig. 3).
The seismic quality tends to degrade towards the West and also with depth. Consequently the structural architecture in those parts is less constrained. On both sections, the top of the Eocene Kirthar limestones (cf. Fig. 3) is indicated on locations where well or seismic data unambiguously allow for that interpretation or where it is constrained by surface geology (Fig. 8, orange interpretation). The top of the Kirthar limestones is one of the most characteristic features in the seismic data. It is represented by a strong, continuous reflector on top of a package of weaker reflectors with good continuity. In the following section, a brief description of the main structures at the level of the Kirthar limestones  is given and structures in areas of good seismic image quality are analysed.

**4.1 Northern sector**

In the northern sector, the undeformed foreland is marked by  approximately horizontal reflectors (Fig. 8a, point a).
Sub-horizontal seismic reflectors indicate the presence of sedimentary rocks to at least 8 km  depth. A minor anticlinal feature (Fig. 8a, point b) and a more pronounced anticline (Fig. 8a, point d) are separated by a zone of discontinuous reflectors (Fig. 8a, point c), that is interpreted as a fault offsetting the Kirthar limestone.  Some more details of these frontal structures is  depicted in the growth strata imaged on a time-domain seismic section nearby (Fig. 9):
The interpreted growth strata packages "a" and "b" show westward thinning and onlap, thus a pattern of apparent progressive
limb rotation. The seismic interpretation at depth indicate a thrust fault (thrust "1" in Fig. 9 a). There is no additional thin-skinned deformation east of the tip of the wedge documented in confidential seismic data east of the section or on the surface. Therefore, this thrust is interpreted as part of a structural wedge, roofed by a bedding parallel thrust (thrust "2" in Fig. 9 a). Tilted strata below thrust "1" with a slightly westward offset kink axis (white stippled line in Fig. 9a) indicate a potential deeper wedge. Fig. 9 b-d show a possible sequence of deformation events that honours the growth strata pattern and structures
identified on the seismic. By stacking two wedges the deformation front can stay relatively stationary and develop a growth strata package similar to the imaged one. Migrating kink bands likely are not resolved due to low sedimentation rates and potential intervals of erosion. Unlike shown the wedges might also be partial active at the same time, complexly accommodating large scale layer parallel shortening. The youngest thrust short-cuts the wedges (Fig. 9d, fault "3" in Fig. 9a) and deforms the youngest growth strata package ("c" in Fig. 9a).
The relatively good seismic image and nearby well control allows to define the stratigraphic level of the roof thrust (fault "2" in Fig. 9). The thrust has a trend that is parallel to the bedding in the Paleocene
shales (upper Ranikot shales, cf. Fig. 3) just below the thick and competent limestones (Sui Main limestones, cf. Fig. 3), characterized by the low-reflectivity seismic character.   The deeper  segments of thrusts "1" and "3"  are relatively uncertain based on the seismic profiles. However, the Jurassic Chiltan Formation  has been drilled in the hanging wall of thrust "3", indicating that the thrust cuts below the Jurassic.
A syncline is located west of these frontal structures, though it is not imaged on the seismic data due to steeply dipping to overturned beds (Fig. 8a, point e). A large scale anticline with Kirthar limestones on surface level is indicated at point f (Fig. 8a, cf. Fig. 5). The low-reflectivity seismic facies below the Kirthar limestones  is the Eocene Ghazij (Shales) Formation  (Fig. 8a, point g). These shales thicken dramatically from the well on the frontal anticline  in the East (several tens of meters) towards the West (several hundreds of meters, constrained by  outcrop information just west of the Kirthar Escarpment, cf. Fig. 5 and Ahmad et al. 2012). A small scale anticline of higher order  exposes Kirthar limestones on the surface (Fig. 8a, point h). A syncline marks the western end of the seismic line (Fig. 8a point i). The Kirthar Formation crops out also at the Kirthar Escarpment west of the section (Fig. 5 northern sector, at approx. 1150 m above sea level). It is notable that the structural elevation of the Kirthar limestone increases from east to west (approx. 2.5 km in the western syncline, Fig. 8a point i). At the highest outcropping point of Kirthar limestones the difference to the estimated regional level is 5.5 – 6 km with an uncertainty related to the interpreted slope of the regional elevation. A rough  depth-to-detachment analysis conducted with the excess area approach (Epard and Groshong, 1993) on the large scale anticline  suggests an upper detachment depth of 8-10

km. The spread in the predicted detachment is due to high uncertainty in the deeper stratigraphic picks on the seismic and the fact that the Kirthar Formation is not returning to regional elevation in the syncline to the West.

**4.2 Southern sector**

In the southern section, the undeformed foreland (Fig. 8b, point a) shows sub-horizontal reflectivity to at least 8 km depth. To the West a minor flexure (Fig. 8b, point b) is situated underneath a seismic noise zone hiding a thrust fault (Fig. 8b, point c). The anticline above the thrust (Fig. 8b, point d) is the southern along-strike continuation of the anticline on the northern section (Fig. 8a, point d). The syncline towards the West (Fig. 8b, point e) is much broader than its northern equivalent. Sub-horizontal reflectors indicate the presence of sedimentary rocks to at least a depth of 10 km. The frontal structures have been analysed and interpreted. Finally the concluded model is illustrated and tested by running a kinematic forward model (Fig. 10). The interpreted fault geometries as well as a stratigraphic template elaborated from wells and outcrop sections are used for the starting configuration of the model (Fig. 10a). Step 1 follows the sequence elaborated in the northern sector (i.e. wedging before fault-propagation folding cf. Fig. 9) which shows a small fault-bend fold that forms a small triangle structure at the deformation front (Fig. 10b). This triangle structure is cut by a subsequent thrust, forming a fault-propagation fold (Fig. 10c). This step is modelled using the tri-shear implementation in Move software (Midland Valley, 2016, Fig. 10c). There are several parameters that control the shape of the anticline. In detail, more than one solution (combination of parameters) can generate an approximate fit to the given constraints (seismic, well data (not shown) and surface dips), but differences are not significant. The reasonable fit shown in Fig. 10c supports the fault interpretation and the amount of shortening applied to these frontal structures (about 5000 m of horizontal shortening). To the west of the red stippled line in Fig. 10c, the model does not exactly match the seismic image- the interpreted Kirthar Formation is constantly rising until the limestones crop out in a small scale fold (Fig. 8b, point f). West of point f the Kirthar formation is continuously exposed to the Kirthar

Escarpment (just west of the end of the seismic line) at an elevation of around 1850 m above sea level.
Similar to the northern section is the  the Eocene Ghazij shales thicken westward

. The shales are thin (several tens of meters) in the frontal anticline (Fig. 8b, below point d) and thicken towards the West (Fig. 8b, point g). This thickening is  especially pronounced west of the clinoforms (Fig. 8b, point h), which  are interpreted as the carbonate margin of the Sui/Laki Limestones.

On the southern section the structural elevation of the Kirthar limestones  rises constantly from the syncline axis (Fig. 8b, point e) towards the Kirthar Escarpment in the West. The structural elevation gain above regional at the Kirthar Escarpment is more than 6500 m (with uncertainty related to the interpreted regional level, Fig. 8b stippled orange line).

**5. Linking thick-skinned and thin-skinned deformation**

Despite some structural differences between the northern and southern sectors, the common observation is the overall increase of the elevation  of the Kirthar and other formations above the regional from the East to the West. Such an increase in elevation can  be explained by several mechanisms: a) a strong wedge shape of the pre-deformational strata below the Kirthar, b) a thrust/fault to a deeper structural level, c) internal structural thickening of formations below the Kirthar Formation or any combination thereof as these proposed mechanisms are not mutually exclusive.

For several reasons discussed throughout this chapter, we propose that the most likely scenario for driving the structural uplift is a thick-skinned contribution that is probably caused by partial inversion of existing structures linking upwards with suitable detachments in the sedimentary column. The balanced sections of the southern and the northern zone are displayed in Fig. 12 and 14, respectively. The sections honour the seismic interpretation and constraints from the structural modelling and fit the regional context and constraints. Before discussing the sections individually in detail we need to elaborate these constraints and arguments. This includes addressing the following main questions: a) Could a pure thin-skinned (duplex) solution explain the same (regional) pattern?  b) Which are the indications for inversion in contrast to a (non-inversion) basement involved model?

**5.1. Constraints from regional structures (thin- vs. thick-skinned)**

West of the area covered in Figure 5 the topographic and structural elevation remains high (more than 6000m), as indicated by the outcropping of the Jurassic in various folds (Figs. 2, 5). A conceptual regional cross section is displayed in Figure 16g. West of the area covered in Figure 5 the section is mainly based on a low resolution geological map (scale, Bannert et al., 1992). The section tentatively shows some relatively steep thick-skinned faults and gently folded strata above. Due to the limited data and the problems of cross-section orientation and non-plane strain conditions balancing of this regional section is problematic. However, the folds shown in the section accommodate approximately 10% of line length shortening. Based on the balancing results from our own sections where higher resolution data is present, we argue that the actual shortening can be somewhat higher (in the order of 15-20%) and that the difference is due to scale problems as well as unresolved shortening in wedges and other distributed shortening. A conceptual section compares a thick-skinned to a thin-skinned solution (Fig. 17). In both cases the deformation is pinned at the deformation front. The thick-skinned model envisages that the total amount of shortening is accommodated in the contractional structures above the approximately equally shortened basement (Fig. 17a). The thin-skinned model assumes that the basement remains undeformed beneath the duplex structures (Fig. 17b) and has to be shortened towards the hinterland. The thin-skinned duplex solution shown (assuming reasonable stratigraphic thicknesses) does not reach the structural uplift observed (more than 6000m) and has serious balancing problems. Increasing the magnitude of shortening would allow to attain higher structural elevation(s), but increases the balancing issue at the same time. The shortening of the strata above the duplexes would likely require a set of back-thrusts as the plane of the roof-thrust is severely folded and thus not likely a viable slip-plane. No such back-thrusts or other structures that would accommodate the excess shortening are observable on the geological maps. Furthermore, the basement would require to shorten somewhere as well with the same magnitude, which would usually happen by a staircase thrust system towards the hinterland – with the consequence that deeper stratigraphic rocks or basement are uplifted (tentatively shown in Fig. 17b). However, towards the hinterland no such root zone is present (cf. Figs. 2, 16 g). Furthermore, the fold belt as such does not show a prominent surface slope (only 0.5° -1°, Fig. 16h). This could indicated that the transpressive fold belt likely does not represent a critical tapered accretionary wedge (cf. Dahlen et al. 1984, Suppe 2007), although, we do not have good control on the basal angle of a potential wedge. A governing wedge shape is probably necessary to allow sustained basal accretion of duplexes.

Steeper faults, also affecting the basement, do require much less shortening to uplift overlying strata to a high structural elevation (Fig. 17 a) and are consistent with the missing root zone towards the plate boundary (Figs. 2, 16 g).

In our study area, Tthe outcropping structures west of the seismic coverage/west of the Kirthar Escarpment yield some indications about the structural architecture below the Kirthar and Ghazij Formations (Fig. 5). The area west of the Kirthar Escarpment has a high mean elevation (more than 1000 m above sea level, cf Fig. 16h). The anticlines with Jurassic outcrops represent the structures with the highest elevations in the area (labelled with bold numbers 1 in Fig. 5). In-between are areas where Paleocene (and sparsely also Eocene) rocks are preserved, which represent relative structural lows (labelled with bold numbers 2 in Fig. 5). A further characteristic is the presence of long wavelength folds with several km wavelength (labelled with bold numbers 3 in Fig. 5) and anticlines folds with much smaller wavelengths and higher frequencies (labelled with bold numbers 4 in Fig. 5), indicating a much shallower detachment horizon. The large scale anticlines are usually double plunging and have roughly NNW-SSE to N-S trending axes, but a variety of additional subordinate directions are present as well. A plausible deformation model should be able to explain this complex pattern.

As lined out above, a thick-skinned contribution to the structural elevation is necessary. In a transpressional system we would expect a zone of shortening in which the shortening features are striking 45° to parallel to the dominant strike-slip features (Sanderson and Marchini, 1984, Fossen et al., 1994, Schreurs and Colleta, 1998). The Chaman and Ghazeraband faults are in an N-S to NNE to SSW orientation (Fig. 1), thus, shortening structures should have a strike orientation of NE-SW to NNE-SSW. The NNW-SSE to N-S trending axes west of the Kirthar Escarpment in the central Kirthar fold belt (Fig. 5) seem rather unusual in respect to the orientation of the transpressive margin. Thus, it seems reasonable to assume, that the NNW-SSE to N-S orientation is not linked to newly initiated faults at depth but is associated to localized deformation controlled by inherited zones of weaknesses.

**5.1 2 Focal mechanism from theConstraints from seismicity southern western fold belt**

We use ISC bulletin database derived nodal planes to constrain potential fault geometries in the subsurface in the wider study area (Fig. 2). Given the tectonic setting of a lateral collision zone, it is not surprising that earthquakes towards the current plate boundary at the Chaman Fault document dominantly strike-slip faulting. Some focal mechanisms of earthquakes close to the deformation front show dominant dip-slip shortening (Fig. 2 and Table 1). All these events are in depth ranges of 10-15 km, with the exception of F4 (Fig. 2 and Table 1), which is at greater depth (33 km). Interestingly is that this event is the only one with a slight oblique character. Given the potential error ranges on the depth of the events the shallow events could be located in the crystalline basement or in the lower part of the sedimentary column.

It is not clear which of the two nodal planes was the moving plane. We could either assume it is the one with the lowest dip or it is westward dipping corresponding to the South-east or eastward directed shortening. For the first assumption fault dips are between 15° - 45° and for eastward dipping faults between 15° - 57°. In both cases, the steeper faults are considered to be too steep to represent newly initiated reverse/thrust faults. We interpret these steep faults therefore as parts of reactivated pre-existing faults that are in a suitable angle for reactivation. The shallower dipping events could represent newly initiated faults, of which those in depth above 12 km could be located in the sedimentary column.

Based on levelling data surface deformation associated with the 1931 Mach Earthquake in front of the northern Kirthar ranges has been investigated by Ambraseys and Bilham (2003) as well as by Szeliga et al. (2009). The authors model different fault slip solutions to match the seismic and post seismic elevation gain at the deformation front. In a geological section Szeliga et al. (2009) consider listric thrust faults with angles exceeding 45° linking shortening on a deep flat decollement (likely in the basement) to higher levels in the sediments (Fig. 16d). In order to match the surface deformation after the 1931 Mach earthquake with elastic models a fault geometry comprising deep detachment, a ramp section (part of a steep listric fault) and a branching gently dipping thrust towards the deformation front are needed. The fault shape considered responsible for the event by Szeliga et al. (2009) is tentatively shown in Figure 16d as red line. Such a geometry supports our proposed model close to the deformation front, where basement faults link with shallower detachments and thrusts in the sediments.

**5.3 A simplified thick-skinned - thin-skinned inversion model**

We propose that an inversion model is the best solution to explain all the different observations and constraints. Yamada and McClay (2004) demonstrated that the shape of the normal fault and associated (half-) graben with its syn-kinematic fill (Fig.

11a) defines the shape of the inversion anticline (Fig. 11b). Interesting to note is the presence of double plunging anticlines and the possibility of local lows in-between the anticlines and towards the hinterland (Fig. 11b), which can be compared toresembles the structural pattern west of the Kirthar Escarpment in map view. The analogue experiments are limited by the rigid boundary conditions (rigid and non-deformable footwall, constant length of the hanging-wall fault-) whereas in nature, this is likely not the case. Inversion likely does affect only those parts of the faults that are suitable for frictional reactivation.

In listric fault systems these would be dominantly the lower/deeper segments located in the ridged basement (depending on the post-rift strain history). Other inversion related deformation like hanging wall shortcut faults, reverse faults, buttressing effects (cf. Cooper et al., 1989; Hayward and Graham, 1989) are likely to be present as well and can be considered as indirect inversion. In the study area the shortening of the inverting normal fault is considered to be transferred to a detachment in the sediments (Fig. 11c), explaining the presence of short wavelength folds adjacent to the large wavelength folds (cf. Fig. 5). The linking from the deeper inverting fault to the detachment in the sediments might be associated with the above mentioned complex deformation, e.g. a footwall short-cut fault (Fig. 11c).

We further suggest that the complex map pattern is likely the result of a much more complex inverted fault pattern as observed in natural rifts. En-echelon pattern and overlaps of faults with intact and broken relay ramps, horses etc. (Fig. 11d) could contribute to a more complex deformation pattern if directly or indirectly inverted. Additionally, several stacked detachment horizons allow to accommodate shortening by linking stacked wedges and distributed ductile strain. As a consequence the amount of shortening introduced by basement faults is partly disseminated and thrusts, if they reach the surface have relatively small displacements (relative to the amount introduced by the basement faults) without the necessity of major thrust faults breaking the surface. We consider that the inverting faults are inherited from the original rift phase on the lateral boundary when the Indian plate rifted from northern Gondwana (Fig. 4a). The direction of the rift faults thus would also define the N-S

direction of the anticlines, which is strongly oblique to the plate kinematic vector.

**5.4 Southern section kinematic model and balanced section**

The kinematical model of the frontal deformation structures in the southern section (Fig. 10) accommodates approximately 5000 m of shortening. However, this amount of shortening is not enough to explain the 6500 m of regional uplift towards the hinterland at the Kirthar Escarpment when taking into consideration reasonable fault dips. A fault with a 45° angle and a displacement of 5000 m would generate a structural uplift of 5000 m from a simple geometric perspective. Either a much steeper fault is necessary (i.e. >52°), or some additional shortening above/in frontassociated to the inverting fault are is required to explain the 6500 m of regional uplift. From careful seismic interpretation and dip-analysis we have interpreted the presence of small passive roof duplexes underneath the soft Eocene Ghazij shales. Below point g in (Fig. 8b) some strong but laterally discontinuous reflector packages are present (between points h and i). The reflectors are interpreted to represent the Paleocene limestones. The discontinuous pattern is interpreted to be caused by poor imaging and by structural imbrication. East of point i (Fig. 8b), a small back-thrust is interpreted. The structural solution is presented in the balanced section (Fig. 12a). By adding the shortening of these small passive roof duplexes to the total displacement on the basement fault, the required regional uplift at the Kirthar Escarpment can be achieved, as is demonstrated in Fig. 13.

To test our model, we generated a The simplified kinematical forward model that shows the evolution from the restored section (Fig. 12b) to the present stage (Fig. 12a). The initial configuration of the model has starting from one major normal fault (Fig. 13a) as in the restoration. For simplicity we will assume syn-kinematical growth in the lower Jurassic formations, although the fault might have been active as normal fault earlier and later as well (see Section 2.2 Tectonostratigraphic evolution section). The frontal triangle and the small identified interpreted back-thrust (corresponding to the wedge modelled in Fig. 10 and the back-thrust east of point i (Fig. 8b) are likely formedis suggested to have formed in an early deformation phase of dominant wedging and layer parallel shortening in the section. In the model, this deformation is linked to slight inversion of the displayed normal fault (Fig. 13b), however, the deformation could also be linked to shortening further in the West which is transferred via thin-skinned detachments. The generation of the interpreted small passive roof thrusts is considered to be the result The of the main inversion of the normal fault  generatesgenerating shortcut faults with a slightly smaller dip angle in the sediments of the footwall. The presence of several weak stratigraphic units allows some wedging as well as the passive roof backthrust in the Ghazij shales (Fig. 13b and c). This stage reflects large scale layer parallel shortening of the stratigraphy above/in front of the inverting normal fault. The youngest deformation is occurring on the thrust in the frontal anticline. In order to explain this we suppose Withthat with increasing inversion above the null point, the pressure stress on the basement in the footwall likely increases and it finally yields. A basement shortcut develops and links with a suitable detachment generating the frontal anticline (Fig. 13d). With such a kinematical model all The the features/constraints visible in the seismic and at surface, and especially, the regional elevationstructural elevation uplift at the Kirthar Escarpment can all be explained with one major fault inversion (Fig. 13 D). West of the Kirthar Escarpment the regional structural elevation remains relatively high and is not dropping as in the simplified model (Fig. 13d hatched area). For that area, additional shortening is required to maintain the high structural elevation, which could be related to associated toadditional (partly) inverting normal faults is required, similar to the sketch section in Fig. 16 g.

[revised manuscript text omitted]

In order to overcome limitations from single deterministic geometries, Butler et al. (2018) propose good documentation, alternative models and to embrace the uncertainties. In this work, we show the original seismic data, review in detail the regional to local context and use these as arguments why we think our presented deformation model is the most plausible for the central Kirthar Fold Belt from the other investigated alternatives. We do not show alternative models, but we highlight our workflow, the considered constraints and indicate uncertainties of the sections. The contribution of deep founded faults with associated thin-skinned deformation can be considered as reliable, and the pure thin-skinned deformation style can be considered as obsolete.

Based on several observations (folding pattern, fold orientation, focal mechanism) this thick-skinned deformation is interpreted to invert inherited zones of weakness from the rift phase that generated the lateral margin of India. This model is very likely, but remains a conclusion, rather than a direct observation.

In detail the interpreted and constructed sections are as good as the constraints allow and thus still have several solutions. The amount of uncertainty in the sections depends also on the level of observation. How much change on our model is a new solution or just a modification is a matter of definition and also scale dependent.

**7. Conclusions**

[revised manuscript text omitted]

International Seismological Centre, Internatl. Seismol. Cent., Thatcham, United Kingdom, On-line Bulletin,
http://www.isc.ac.uk, 2015

Jadoon, I. A. K., Lawrence, R. D., and Lillie, R. J.: Balanced and retrodeformed geological cross-section from the frontal Sulaiman Lobe, Pakistan: Duplex development in thick strata along the western margin of the Indian plate, in: Thrust tectonics and hydrocarbon systems, edited by: K. R. McClay, AAPG Memoir, 82, 343-356, 1992

Jackson, J.A., Reactivation of basement faults and crustal shortening in orogenic belts, Nature, 283, 343–346, 1980

Jadoon, I.A.K.; Lawrence, R.D.; Lillie, R.J.: Evolution of foreland structures: an example from the Sulaiman thrust lobe of Pakistan, southwest of the Himalayas, Geol. Soc. Spec. Publ., 74, 589-602, 1993

Jarvis A., Reuter, H. I., Nelson, A., and Guevara, E.: Hole-filled seamless SRTM data V4, International Centre for Tropical Agriculture (CIAT), http://srtm.csi.cgiar.org, 2008

Kadri, I.B.: Petroleum Geology of Pakistan. Pakistan Petroleum Limited, Karachi, Pakistan, 275 pp., 1995

Khan, S. D.. Walker, D. J., Hall, S. A., Burke. K. C. Shah, M. T. and Stockli, L.: Did the Kohistan-Ladakh island arc collide first with India?, GSA Bulletin, 121, 366–384, 2009

Kufner, S-K., Schurr,, B., Sippl, C., Yuan, X., Ratschbacher, L., Akbar, M., Ischuk, A., Murodkulov, S., Schneider, F., Mechie, J., and F. Tilmann, F.: Deep India meets deep Asia: Lithospheric indentation, delamination and break-off under Pamir and

Hindu Kush (Central Asia). Earth Planet. Sc. Lett., 435, 171–184, doi.org/10.1016/j.epsl.2015.11.046, 2016.

Lacombe, O. and Bellahsen, N.: Thick-skinned tectonics and basement-involved fold–thrust belts: insights from selected Cenozoic orogens, Geol. Mag., doi:10.1017/S0016756816000078, 2016.

Lawrence, R.D., Yeats, R.S., Khan, S.H., Farah, A., and DeJong, K.A.: Thrust and strike slip fault interaction along the Chaman transform zone, Pakistan, Geol. Soc. Spec. Publ., 9, 363-370, 1981.

Lentas, K., Di Giacomo, D., Harris, J., and Storchak, D.: The ISC Bulletin as a comprehensive source of earthquake source mechanisms, Earth Syst. Sci. Data Discuss., https://doi.org/10.5194/essd-2018-143, in review, 2018

Mahoney, L., Hill, K., McLaren, S., and Hanani, A.: Complex fold and thrust belt structural styles: Examples from the Greater Juha area of the Papuan Fold and Thrust Belt, Papua New Guinea, J. Struct. Geol., 100, 98-119, DOI: 10.1016/j.jsg.2017.05.010, 2017.

Midland Valley 2016, Move Software, Midland Valley Exploration Ltd, Glasgow, UK, 2016.

Mohadjer, S., Bendick, R., Ischuk, A., Kuzikov, S., Kostuk, A., Saydullaev, U., Lodi, S., Kakar, D.M.,Wasy, A., Khan, M.A., Molnar, P., Bilham, R., and Zubovich, A.V.: Partitioning of India–Eurasia convergence in the Pamir-Hindu Kush from GPS measurements, Geophys. Res. Lett., 37, http://dx.doi.org/10.1029/2009GL041737, 2010

Nemčok, M., Mora, A. and Cosgrove, J.: Thick-skin-dominated orogens; from initial inversion to full accretion: an introduction. In: Thick-Skin-Dominated Orogens: From Initial Inversion to Full Accretion, edited by Nemčok, M., Mora A. and Cosgrove j.), Geol. Soc. Spec. Publ., 377, 1-17, 2013.

Pérez-Gussinyé M., Reston T. J.: Rheological evolution during extension at passive non-volcanic margins: onset of serpentinization and development of detachments to continental break-up, J. Geophys. Res., 106, 3691–3975, 2001.

Reston, T. and Manatschal, G.: Rifted margins: building blocks of later collision. In Arc-Continent Collision: Frontiers in

Earth Sciences, edited by: Brown, D. and Ryan, P.D., Springer-Verlag, Berlin: 3–21, 2011.

Reynolds, K., Copley, A., and Hussain, E.: Evolution and dynamics of a fold-thrust belt: the Sulaiman Range of Pakistan, Geophys. J. Int., 201, 683–710, 2015.

Sanderson, D. and Marchini, R.D., Transpression. J. Struct. Geol., 6: 449–458, 1984

Schelling, D. D.: Frontal structural geometries and detachment tectonics of the northeastern Karachi arc, southern Kirthar Range, Pakistan, in Himalaya and Tibet: Mountain Roots to Mountain Tops, edited by Macfarlane, A., Sorkhabi, R. B. and Quade, J., Geol. S. Am. S., 328, 287-302, DOI: 10.1130/0-8137-2328-0.287, 1999.

Schreurs, G., Colleta, B., Analogue modelling of faulting in zones of continental transpression and transtension. In: Continental Transpressional and Transtensional Tectonics, edited by:  Holdsworth, R.E., Strachan, R.A. and Dewey, J.F., Geol. Soc. Spec. Publ., 195. 59-79, 1998

Scotese, C.R.: The PALEOMAP Project PaleoAtlas for ArcGIS, version 1, Volume 3, Triassic and Jurassic Paleogeographic and Plate Tectonic Reconstructions, Maps 32 – 48, PALEOMAP Project, Evanston, IL, DOI:10.13140/RG.2.1.4108.5685, 2014a.

Scotese, C.R.: The PALEOMAP Project PaleoAtlas for ArcGIS, version 1, Volume 2, Cretaceous Paleogeographic and Plate Tectonic Reconstructions, Maps 16 – 31, PALEOMAP Project, Evanston, IL, DOI:10.13140/RG.2.1.2011.4162, 2014b.

Scotese, C.R.: The PALEOMAP Project PaleoAtlas for ArcGIS, version 1, Volume 1, Cenozoic Paleogeographic and Plate Tectonic Reconstructions, Maps 1 – 15. PALEOMAP Project, Evanston, IL, DOI:10.13140/RG.2.1.2535.7041, 2014c.

Scotese, C.R.: PALEOMAP PaleoAtlas for GPlates and the PaleoData Plotter Program, PALEOMAP Project,
http://www.earthbyte.org/paleomap-paleoatlas-for-gplates, 2016.

Smewing, J. D., Warburton, J., Cernuschi, A., and Ul-Haq, N.: Structural inheritance in the southern Kirthar fold belt: Soc. Petrol. Eng.- Pakistan Association of Petroleum Geologists, 26–34, 2002a.

Smewing, J.D., Warburton, J., Daley, T., Copestake. P., and Ul-Haq, N.: Sequence stratigraphy of the southern Kirthar Fold Belt and Middle Indus Basin, Pakistan, in The Tectonic and Climatic Evolution of the Arabian Sea Region, edited by: Clift,
P.D., Kroon, D., Gaedicke, C. and Craik, J., Geol. Soc. Spec. Publ., 195, 273–299, 2002b.

Smith A. G.: A review of the Ediacaran to Early Cambrian ('Infra-Cambrian') evaporites and associated sediments of the Middle East, Geol. Soc. Spec. Publ., 366, 28, 2012.

Suppe, J., Absolute fault and crustal strength from wedge tapers. Geology, v. 35; no. 12; p. 1127–1130, 2007

Szeliga, W., Bilham, R., Schelling, D., Kakar, D.M., and Lodi, S.: Fold and thrust partitioning in a contracting fold belt:
insights from the 1931 Mach earthquake in Baluchistan. Tectonics, 28, http://dx.doi.org/10.1029/2008TC002265, 2009.

Tectostrat: Mehar Block Study 2000/2001, International Tectostrat Geoconsultants B.V.: un-published report, 2001.

Wandrey, C. J., Law, B.E., and Shah, H.A.: Sembar-Goru/Ghazij Composite Total Petroleum System, Indus and Sulaiman-Kirthar Geologic Provinces, Pakistan and India, in: Petroleum Systems and Related Geologic Studies in Region 8, South Asia, edited by: Wandrey, C. J., U.S. Geological Survey Bulletin, 2208-C, 23 pp., 2004.

Yamada, Y., and McClay, K.: 3-D Analog modeling of inversion thrust structures, in: Thrust tectonics and hydrocarbon systems, edited by: K. R. McClay, AAPG Memoir, 82, 276-301, 2004.

Zaigham, N.A., and Mallick, K.A.: Prospect of Hydrocarbon Associated with Fossil-Rift Structures of the Southern Indus Basin, Pakistan, AAPG Bull., 84, 1833-1848, 2000.

**Figures and captions**

[Figure]

Figure replaced

Figure 1: Simplified structural sketch of the wider Kirthar fold-belt area on a shaded relief map. Location is indicated in in the inset map. The approximate plate motion is from Mohadjer et al. (2010); Locations of sSections in Fig. 16 from other studies are indicated: a: Jadoon et al. (1992), a+bb+c: Banks and Warburton (1986), c: Jadoon et al. (1992), d:Szeliga et al. (2009), e: Fowler et al. (2004), ef: Schelling (1999), g: this study..

**Figure replaced**

**Figure 2: Nodal planes from International Seismological Centre (2015)** database  DEM with draped geological map after Bannert et al. (1992) and selected structural elements. **Location of the figure is indicated in Fig. 1.** Dotted red lines are locations of sections in Fig. 16. **Labelled events are listed in Table 1.** Yellow star marks the 1931 Mach event after Szeliga et al. (2009).

[Figure]

**Figure replaced: added seismic horizons**

**Figure 3: Litostratigraphic overview with hydrocarbon play elements and mechanical stratigraphic interpretations (after Kadri, 1995; Tectostrat 2001; Smewing et al. 2002b and author observations). S.H. = seismic horizons used in this study**

[Figure]

**Figure 4: Paleogeographic evolution of the study area as part of the Indian plate since the Jurassic. (a) Jurassic, ca. 175 million years, a rift evolves northwest of the approximate study area location (star; map from Scotese, 2014a), (b) ca. 65 million years, Cretaceous/Paleocene, drifting northward. An  Island arc (likely an intra oceanic arc) is visible north of the approx. study area location (map from Scotese, 2014b). (c) ca. 50 Eocene, post ophiolite obduction, but pre-collision with Eurasia (map from Scotese, 2014c). (d) ca. 20 Miocene, early collision stage with flexural foreland stage (map from Scotese, 2014c).**

[Figure]

**Figure 5: Semi-transparent lithostratigraphic map (modified after Tectostrat, 2001) of the study area draped in Google Earth. Dotted rectangles with labels a, b, c indicate approx. areas seen in slanted view in Fig. 6a-c. Red labels a-d indicate locations of field photographs in Fig. 7. Northern and southern sectors of the fold belt are indicated by bold red and green lines, respectively. Bold white numbers indicate examples for 1: structural highs, 2: structural lows, 3: long wavelength anticline, 4: short wavelength folding.**
**K.E:= Kirthar Escarpment**

[Figure]

**Figure 6: Example for remote field work with Google Earth. Locations of the views are indicated in Fig. 5. (a) Recent mass wasting: Blocks of Kirthar Limestones glide down the eroded flanks of soft Ghazij Formation; (b) Sub-recent to recent mass wasting: a large slab of Kirthar Limestones from the anticline roof is now folded over the  forelimb of the anticline _(cf. Supplementary Figure I)_. In the background extensional faults are visible _in the Kirthar limestones_  _representing_ the roof of the  anticline . The limestones partly glide_/collapse_ over the vertical beds of the forelimb. (c) Disharmonic folding: Jurassic rocks show large wavelength folding, while the hard limestones of the Cretaceous Parh formation are folded in smaller wavelength and higher frequency. A weak decollement zone is located in the Goru Shales.**

[Figure]

**Figure 7: Examples for observations from fieldwork. (a) Sub-recent conglomeratic sediments are folded and eroded. (b) Striations on a bedding plane in (Pleistocene?) conglomerates indicating flexural slip folding. (c) small scale anticline in Nari Formation rocks. The amplitude and wavelength of the fold suggest, that the lower detachment horizon is likely in lower Nari Formation. (d) small to medium scale folding in Kirthar and Nari Formations. The fold is a mappable feature (cf. Fig. 5) and indicates to a detachment horizon below the Kirthar limestones.**

[Figure]

Figure replaced, added well control and horizon information

**Figure 8: Two W-E composed seismic sections of pre-stack depth migrated seismic. Orange interpretations indicate clearly constrained Top Kirthar Formation from seismic, wells and outcrop. Stippled orange line is anticipated pre-contractional regional elevation of the Kirthar Formation. K.E.: Kirthar Escarpment. A high resolution image without interpretation is available as supplementary figure. The final interpreted seismic lines are part of Figs. 12a and 14a; (a) Seismic section composed from two overlapping 2D seismic lines in the northern Sector (exact position not shown for confidentiality reasons). W1: well control within 4 km to Jurassic level. W2: well control within 5 km to Paleogene level. Labels a-i are used to indicated features discussed in the text. Numbered horizons in the East refer to horizons as in Fig. 3 (b) section composed from 2D and 3D seismic data. W3: well control within 1 km to Upper Cretaceous level. Labels a-i are used to indicated features discussed in the text. Numbered horizons in the East refer to horizons as in Fig. 3**

[Figure]

**Figure replaced**

**Figure 9: (a) 2D seismic section in time domain in the northern sector (exact position not shown for confidentiality reasons) with fault and growth strata interpretation. (b-d) show a possible solution for the growth strata pattern as discussed in the text. Seismic in the background of (d) is roughly depth converted version of (a).**

[Figure]

**Figure 10: Forward modelling of the frontal structures in the southern sector (a) The seismic image, surface geometry including dips, the present day deformed state of the Top Eocene limestones (constrained by nearby well control) as well as the interpreted and constructed faults are given as reference frame for the forward model. The model uses a stratigraphic wedge with thicknesses which are constrained by well and outcrop observations. (b) A small triangle structure at the deformation front is modelled with fault-bend folding. The lower detachment (1) is in lower Jurassic or Triassic succession, the upper one is interpreted in the soft Cretaceous Goru shales (dotted line). (c) A fault-propagation fold forms hinterland-ward of the triangle structure by a thrust ramp (2) modelling done with tri-shear. The model mimics the structure imaged in the seismic approximately from the deformation front (east) to the red stippled line.**

[Figure]

**Figure replaced**

Figure 11: (a) and (b) Model of extension with subsequent inversion on curved linked faults (modified after Yamada and McClay, 2004). (c) Adding a-thin-skinned deformation and short-cut  fault to the sketch of inverted curved linked fault system (d) Sketch of half graben systems with overlapping faults for anticipation of more complex subsurface geometries before inversion.

[Figure]

**Figure 12: (a) Constructed W-E section in the southern sector of the study area with PSDM seismic in the background (i.e. Figure 8b). The section is balanced between the red lose line and the eastern end of the section (fixed line), KE: Kirthar Escarpment; (b) restored section (50% scale of (a)). Calculated shortening is approx. 10 km or 20%.**

[Figure]

**Figure 13: Simplified kinematical evolution of the southern sector. (a) Pre-contractional situation with Jurassic normal fault. Thin stippled lines indicate faults of dominant layer parallel shortening, (c) and (b) Incremental deformation of imbrication and passive roof thrusting above the inverting normal fault. (d) Final geometry of the kinematical forward model compared to seismic and surface geology. The geometry in the hatched area in the western part of the section does not fit the surface geology and would require additional deformation by inverting faults and cover sediments not regarded in this model.**

[Figure]

**Figure 14: (a) Constructed W-E section in the northern sector of the study area with PSDM seismic in the background (i.e. Fig. 8a). Dip-measurements projected between 2.5 and 4km. The section is balanced between the red lose line and the eastern end of the section (fixed line). (b) Restored section (at 50% scale of (a)) by using line length and area balancing methods. Calculated shortening is approx. 11.2 km or 18%**

[Figure]

**Figure 15: Simplified kinematical evolution of the southern sector, shown for Eocene to Basement rocks. (a) pre-contractional situation with Jurassic normal faults. Stippled line indicated future shortcut fault. (b) and (c) Increments of inversion with shortcut faulting and detachment folding related to buttressing. (d) Final geometry of the kinematical forward model.**

[Figure]

**Figure 16 Selected sections in southern western fold belt of Pakistan (for section locations see Figs.1 and 2). a: after Jadoon et al. (1992), b+c: after Banks and Warburton (1986), d:after Szeliga et al. (2009), red fault indicate approx. shape of fault considered responsible for Mach 1931 event, e: after Fowler et al. (2004), f: after Schelling (1999), g: this study, the frontal part (strong colours) is a slightly projected section elaborated for the northern sector of this study area. The western part is a tentative regional sketch section based on the geological map of Bannert et al. 1992, as shown in Fig. 2, G.F. = Ghazeraband fault, h:average topography of a 20 km wide section centred on the trace of (g). 5 times vertical exaggeration.**

[Figure]

**Figure 17 simplified sketch comparing thick- vs. thin skinned solutions for the structural elevation uplift of the Jurassic level. Elevations are in respect to the regional elevation of the Jurassic (0 km); a) A series of thick skinned faults with a total of 20 km shortening (16%). Due to the pinning at the deformation front a roof thrust under the horizons need to be present. The excess line length of the sediments above the roof thrust needs to be accommodated in the section as well, which could happen in internal shortening and amplification of the folds. The solution explains a structural uplift towards the hinterland of 5-7 km. b) One example of a duplex solution with total 48 km of shortening (38%). Due to the pinning at the deformation front a roof thrust under the horizons need to be present. The excess line length of the sediments above the roof thrust (about 40km) needs to be accommodated in the section as well. The example solution stays below 5 km structural uplift. In order to increase the uplift, more shortening in the duplexes would be required, which would increase the balancing issues. A tentative thrust cutting into the basement behind the duplexes would uplift basement rocks towards the hinterland.**

| Figure Label | Event Id | Mw | Z | Date | Author | Dip-azimuth | Dip | Strike | Rake | Dip-azimuth | Dip | Strike | Rake |
|---|---|---|---|---|---|---|---|---|---|---|---|---|---|
| F1a | 603867342 | 5.4 | -10100 | 19.12.2013 | NEIC | 253.8 | 14.7 | 163.8 | 70.9 | 93.5 | 76.2 | 3.5 | 94.9 |
| F1b | 603867342 | 5.4 | -12000 | 19.12.2013 | GCMT | 291.0 | 39.0 | 201.0 | 133.0 | 61.0 | 63.0 | 331.0 | 61.0 |
| F1c | 603867342 | 5.4 | -12000 | 19.12.2013 | NEIC | 277.0 | 26.0 | 187.0 | 99.0 | 87.0 | 64.0 | 357.0 | 85.0 |
| F2 | 308027 | 5.4 | -15000 | 21.01.1992 | HRVD | 306.0 | 48.0 | 216.0 | 104.0 | 105.0 | 44.0 | 15.0 | 75.0 |
| F3 | 301671 | 4.9 | -15000 | 28.03.1992 | HRVD | 272.0 | 57.0 | 182.0 | 89.0 | 93.0 | 33.0 | 3.0 | 91.0 |
| F4 | 259589 | 5.1 | -33000 | 28.12.1992 | HRVD | 263.0 | 33.0 | 173.0 | 32.0 | 146.0 | 73.0 | 56.0 | 119.0 |
| F5 | 13436558 | 5.0 | -12000 | 17.03.2009 | GCMT | 304.0 | 45.0 | 214.0 | 106.0 | 102.0 | 47.0 | 12.0 | 75.0 |
| F6 | 604543379 | 5.0 | -12000 | 08.05.2014 | GCMT | 212.0 | 42.0 | 122.0 | 69.0 | 59.0 | 52.0 | 329.0 | 108.0 |

**Table 1: Nodal planes from International Seismological Centre (2015) database, reviewed events only. In addition to the dip the dip-azimuth of the planes is calculated. The strike and rake values from the database are given for completeness and assessment of obliquity. Event F1 has 3 different solutions in the database. The differences are a rough indication of the uncertainty of the data. Green/red dip values indicate the lower and higher dip surfaces of the pair. The author column refers to the original provider in the database (cf. Lentas et al.; 2018 and references therein).**

**Supplementary figures on the following pages/extra PDF**

[Figure]

Supplementary Fig. 1

Schematic evolution showing land-slide interpretation of Fig. 6b.
a) Erosion affects roof of anticline
b) Gravitational induced ductile deformation of soft shales leading to formation of "roof and wall" structure (knee), as suggested by Harrison and Falcon (1934, 1936) for several anticlines in Iran.
c) Anticipated present day situation after the structure has collapsed into a "slip sheet"  (Harrison and Falcon 1934, 1936)
d) Alternative starting geometry with could lead to similar solution

[Figure]

Supplementary Fig. 2

[Figure]

Supplementary Fig. 3

---

## Author Comment (AC3) · 20 Feb 2019

**Reply to RC2 by Referee #2 Jonas Kley**

We thank the reviewer for the effort of the thorough review of the manuscript. We think that our revisions based on the comments significantly improved the revised manuscript.

Our responses to the specific comments of RC2:

**1) RC2**
Page C2 Paragraph 1): "In the way of data, the weakest part of the paper is definitely the claim that reactivated normal faults are involved in the deep structure. The seismic profiles do not reach deep enough to show anything conclusive. The earthquake nodal planes except one at 57° dip too gently to satisfy the Coulomb-Mohr prediction for normal faults. In fact, the average (arithmetic mean) dip angle of the west-dipping nodal planes is only 38°, much closer to an ideal Mohr-Coulomb thrust fault than normal fault. The normal faults of the structural model dip around 50°. Judging from the stratigraphic description and the authors' comments, the timing of active rifting isn't very well constrained, either. The same seems to hold true for the depth to and nature of the basement. I therefore recommend to tone down the inversion-related part of the interpretation while maintaining that the basement must be involved in thrusting".

**Response:**
The interpretation that the required basement involvement in the deformation is related to inversion is not only based on the fact that some of the nodal planes are too steep to represent newly initiated faults but on a combined series of observations. We will elaborated this below and have changed Section 5 accordingly.

Jonas Kley suggest that the average of the west-dipping nodal planes (38°) is close to the ideal angle for Mohr-Coulomb faults. We consider that taking an average is not suitable for this discussion. In our manuscript and in our structural model we suggest that only the steeper events are considered to represent inverting normal faults. In our model there are also newly initiated thrust faults (flats and ramps) in the sedimentary succession, which would also contribute to the events recorded. An average value thus is not indicative.

On page 9 line 25 of RC2 supplement it is commented that steeper faults are also "too gently dipping to represent normal faults".

We do not fully agree to this point. The angles are below the 60° usually assumed for the formation of normal faults. The listric nature of extensional faults in rift systems can lead to rotation of faults in the hanging wall, making them more suitable for reactivation (as proposed for example by Jackson 1980). Direct fault inversion indeed is a complex matter. Fault angles, partly rotated through the extensional history, as wells fluid pressures on inversion may play a role (Sibson, 1995). It is beyond the scope of this manuscript to discuss all the potential parameters - instead, we changed the text to clarify, that we consider the presence of inversion as a most likely scenario based on a line of constraints (reworked Section 5). We clarify that we do not only consider (partly) fault inversion, but also other deformation structures often associated with inversion, where faults are not suitable for direct inversion (e.g. hanging wall shortcuts, buttressing effects, back-thrusts, cf. Cooper et al., 1989; Hayward and Graham, 1989).

We expanded Section 5.1 to explain the regional constraints which we interpret as being indicative of inversion rather than non-inversion basement thrusting. The main argument is the orientation of the fold axes west of the Kirthar Escarpment (NNW-SSE to N-S) which does not fit well with shortening one would expect from transpression with the main left lateral strike-slip faults striking N-S to NNE to SSW.

With all the clarifications given above (and implemented in the MS) we think it is justified to keep our interpretation that the deformation in the study area is related to inversion.

Nevertheless, we clearly state that this model remains an indirect conclusion on several observations and interpretations, rather than a direct observation (6.5 Uncertainty).

**Changes on the MS:**
Section 5 and subsection have been re-ordered and extended to improve the line of argumentation based on all constraints. This includes addressing the following main questions: a) could a pure thin-skinned (duplex) solution explain the same (regional) pattern? (requested by RC1) b) Which are the indications for inversion in contrast to a (non-inversion) basement involved model?

All constraints from geological maps (Fig. 5 and reworked Fig. 2) and the new regional section (new Fig. 16g) are elaborated in subsection 5.1 "Constraints from regional structures". The former sub-section "Deformation pattern west of the Kirthar Escarpment is included here). After this, the constraints from the focal mechanisms are elaborated (Section 5.2.). To the sub-section 5.3 "A simplified thick-skinned - thin-skinned inversion model" we added that we consider also other inversion features and a higher complexity (Cooper et al., 1989; Hayward and Graham, 1989). See text as in the track-changes manuscript file.

Section 6.5 Uncertainty, added: "The inversion model is very likely, but remains a conclusion, rather than a direct observation".

**2) RC2**
Page C2 Paragraph 2): "When the authors compare their new structural models to Banks and Warburtons passive-roof duplex interpretation they should at least briefly discuss what happens in the more internal parts of the belt, away from the deformation front. The passive roof model was motivated by the need to explain gently folded strata raised well above the regional level for a considerable across-strike distance. I assume that this problem also applies to the central Kirthar Range. If the Kirthar Range is held up by a series of reactivated normal faults, where is the reverse displacement of the more internal faults accommodated that cannot be transferred to the thin-skinned front? Or, in other words, is there enough shortening in the internal Kirthar Range to support its topographic and structural elevation assuming that the basal detachment is in the basement?"

**Response:**
The question raised is interesting, but difficult to answer. We do not have enough geological data to confidently constrain the amount of shortening all across the lateral mountain belt towards the strike slip faults at the plate boundary. West of our core study area we do not have a high resolution geological map, no bedding dip information and no thickness control on the stratigraphy etc. However we address this question as best as possible in Section 5.1, making use of the new regional sketch section Fig. 16 g (as recommended by the referee, see Point 5 below) as well as a small sketch to explain the consequences of thick-skinned/thin-skinned shortening (Fig. 17).

The regional sketch section indicates a deformation style with some inversion and distributed ductile deformation (including wedging/LPS), similar to what we observe in the study area in the front of the section. The amount of shortening in the gently folded strata west of our study area has only about 10% shortening (very rough estimated as the section is partly oblique to some of the fold axes. Furthermore, there are no plane strain conditions in a regional section in a transpression zone). Due to the poor input data, we assume that the actual shortening is in the order of 15% if one would include the non-resolved deformation (wedging/LPS, our sections show 18-20% shortening). We consider that this amount of shortening can be locally accommodated and does not need to propagate to the deformation front – However, with the present data at hand, we cannot define this any better. With 15% of shortening a thick-skinned solution for the structural elevation west of the Kirthar Escarpment can be justified. A thin-skinned solution can be ruled out.

**Changes on the MS:**
Section 5.1: Description of the added regional sketch cross-section (Fig. 16g as well as a geological map background in Fig. 2). The regional shortening and how that can be accommodated are discussed (using new Figure 17). See track-changes MS.

**3) RC2**
Page C2 Paragraph 3 "I am not entirely convinced by the uniqueness of the sequence of thrusting derived in Fig. 9. The advance of a thrust wedge between thrust 1 and backthrust 2 would result in kink band migration and not "progressive limb rotation" as described by the authors (l. 22 in text) and actually suggested by the growth strata geometries. It is also interesting that the kink axis shown to be associated with the tip of the wedge at deeper levels appears offset to the west in the growth strata, but also in the syncline suggested below thrust 1. I could imagine a scenario with no bedding-parallel backthrust and thrust 1 as a late subhorizontal structure displacing the syncline axis towards the east. The implication would be that there must be another thin-skinned thrust further east.

**Response:**
Indeed, thrust wedge advance of thrust 2 would lead to kink band migration. The observed strata indicates rather limb rotation (although small internal unconformities in the growth strata could mask a higher complexity. The suggested order of deformation by the Referee can be excluded, as there is no additional thin-skinned deformation east of the tip of the wedge (not in the confidential seismic data and there are no indications on surface geology etc.). However, the referee accurately observed the presence of another, deeper kink, offset in respect to the kink in our original Figure 9. We include this to reinvestigate the potential kinematics. Figure 9 has been extended to show a possible evolution of the frontal system. The forelimb of the anticline apparently stays relative stationary due to the stacking of two wedges. Therefore, the growth strata would build two small stacked wedges with kink-band migration that appear similar to limb rotation in the image. This is not necessarily the only explanation and kinematical order, but one in-line with the observations.

**Changes on the MS:**
Figure 9 has been extended.

In section 4.1 the new Figure 9 and the suggested evolution is described. The caption has been adapted accordingly (see track-changes MS).

**4) RC2**
Page C3 Paragraph 2 upper part "One thing I am deeply skeptical about is the landslide interpreted in Fig. 6 b. The way this feature is described in the caption I gather that it is supposed to have formed by draping over the topography of the steep forelimb (or did I get that wrong?). I find it hard to believe that you could form the orderly anticline depicted in the satellite image from a rock mass sliding over an irregular land surface."

**Response:**
The original explanation in the text of Section 3.1 for Fig. 6b is probably too short to reasonably explain this feature. The explanation has been extended. A model is proposed based on similar features known from the Zagros.

**Changes on the MS:**
We adapted the text and added the reference to similar features in Iran (Harrison and Falcon, 1934, 1936); see track-changes MS. Furthermore, we added a supplementary figure to illustrate a potential evolution of the slide (into supplement, because we consider this not of key importance for the manuscript and the conclusions).

**5) RC2**

Page C3 Paragraph 2 lower part: "I think that the paper would strongly benefit from a few additional figures. First, it would help the imagination to have a regional cross-section reaching west to the strike-slip system. Secondly, I strongly recommend to prepare a synthetic figure that combines the new cross-sections with those from published studies whose locations are shown in Fig. 1, preferrably redrawn such that comparison is made easy. Nobody wants to look up four other papers to see what the paper they are presently reading is talking about"

**Response**

Agreed. We prepared an additional figure (Figure 16, note figure numbers will be resorted in the final revised MS), which shows redrafted sections studies mentioned in Section 1. Additionally we added a tentative regional sketch section (Fig. 16g) which includes in the frontal part the simplified version of Fig. 14a, the final section of the northern sector. The original sections (Fig. 12 and Fig. 14) are in a smaller scale, thus showing more details and therefore should remain as they are. The simplified version in Figure 16g should serve the requested purpose to be able to compare scale and style of our results to the published sections.

**Changes on the MS:**

Added Figure 16 which includes both sections from published studies and a regional section including study results. The background in Fig. 2 has been replaced to show the geological map the regional section is based on. The description of the regional section is now part of the extended Section 5 (as already mentioned above), see track changes MS.

**Comments on figure pages in se-2018-137-RC2-supplement.pdf**

Referee 2 (Jonas Kley) left some comments and text correction suggestions in two supplementary PDFs. As stated in se-2018-137-RC3.pdf the first supplementary PDF (se-2018-137-RC2-supplement.pdf) contains uncommented mark-ups which are removed or commented in the file se-2018-137-RC3-supplement.pdf. Comments to the figures are only in the file se-2018-137-RC2-supplement.pdf, which will be answered below.

An additional reply will be given for RC3 containing the responses to the comments in the to the manuscript text in file se-2018-137-RC3-supplement.pdf

**se-2018-137-RC2-supplement.pdf Page 24, line 4.**

"What is actually visible is a strip of emergent land, presumably the Kohistan-Ladakh arc. It would be nice to add to this figure the locations and kinematics of plate boundaries"

**Response and Change on MS:**

We changed the text in the caption accordingly. Adding locations and kinematics to this figure would indeed be a nice addition but we do not have the database. Researching this seems out-of scope for the purpose of this paper.

**se-2018-137-RC2-supplement.pdf Page 25**

Explain Kirthar Escarpment

**Response and Change on MS:**

Added to the caption

**se-2018-137-RC2-supplement.pdf Page 31**

Several small suggestion to improve Figure 11

**Response and Change on MS:**

The suggestions have been accepted and are implemented in the reworked Figure 11

"Why have you chosen to ignore seismics here?"

**Response**

The seismic in the background is a combination of a 2D line (the western line) and a section from a 3D seismic (as described in caption to Fig. 8). The 2D line shows on both ends acquisition and processing artefacts especially at greater depth with reflectors curving up. The 3D data shows sub-horizontal reflectors at depth. Therefore, the deepest (dipping) reflectors on the eastern end of the 2D seismic line have been considered unreliable and have been ignored.

Mismatch between seismic and dip measurements (Fig. 14)

**Response and Change on MS:**

The dip measurements in the northern sector of the study area were collected by different campaigns before the acquisition of the seismic. Thus, they are not located on the seismic trace and needed projection between 2.5 and 4 km. Additionally, some locations of the older measurements in the northern sector might not be exact (without GPS).

The projection distance has been added to the caption of Fig. 14

**References not listen in the revised MS:**

Sibson, R. H., Selective fault reactivation during basin inversion: potential for fluid redistribution through fault-valve action, Geol. Soc. Spec. Publ., 88, 3-19, 1995

---

## Author Comment (AC4) · 20 Feb 2019

[revised manuscript text omitted]